# Signatures of a surface spin–orbital chiral metal

Federico Mazzola[1,2,13 ✉], Wojciech Brzezicki[3,4,13], Maria Teresa Mercaldo[5], Anita Guarino[6], Chiara Bigi[7], Jill A. Miwa[8], Domenico De Fazio[1], Alberto Crepaldi[9], Jun Fujii[2], Giorgio Rossi[2,10], Pasquale Orgiani[2], Sandeep Kumar Chaluvadi[2], Shyni Punathum Chalil[2], Giancarlo Panaccione[2], Anupam Jana[2], Vincent Polewczyk[2], Ivana Vobornik[2], Changyoung Kim[11], Fabio Miletto-Granozio[12], Rosalba Fittipaldi[6], Carmine Ortix[5], Mario Cuoco[6 ✉] & Antonio Vecchione[6 ✉]

The relation between crystal symmetries, electron correlations and electronic structure steers the formation of a large array of unconventional phases of matter, including magneto-electric loop currents and chiral magnetism[1–6]. The detection of such hidden orders is an important goal in condensed-matter physics. However, until now, non-standard forms of magnetism with chiral electronic ordering have been difficult to detect experimentally[7]. Here we develop a theory for symmetry-broken chiral ground states and propose a methodology based on circularly polarized, spin-selective, angular-resolved photoelectron spectroscopy to study them. We use the archetypal quantum material $Sr_2RuO_4$ and reveal spectroscopic signatures that, despite being subtle, can be reconciled with the formation of spin–orbital chiral currents at the surface of the material[8–10]. As we shed light on these chiral regimes, our findings pave the way for a deeper understanding of ordering phenomena and unconventional magnetism.

A central problem in condensed-matter physics is the existence of unconventional magnetism beyond the usual forms, arising from the long-range order of magnetic dipole moments $\mu = (2S + L)$, where $S$ and $L$ are the electron spin and orbital angular momentum, respectively[11–15]. Such magnetic dipole moments normally arrange spatially with ordered patterns in the crystal, but other forms of magnetic phase may still originate from an electronic ordering resulting from charge currents at the atomic scale. Such phases are odd in time (time-reversal symmetry is broken), inherently subtle and difficult to observe and they are often associated with a hidden magnetic order[1,2,16].

The transport properties of all metallic quantum materials are determined by the spin and orbital degrees of freedom of the Fermi surface. Broken symmetry states (mirror and/or time) with charge currents can have an internal structure with a combination of spin and orbital angular momentum. The spin and orbital angular momentum are pseudovectors with magnetic dipolar nature, so we can refer to their product as a spin–orbital quadrupole or orbital quadrupole. Spin and orbital angular momentum are odd in time and the current is an odd function of a crystal wavevector. Currents carrying spin–orbital or orbital quadrupoles, which are even in time, therefore break time-reversal symmetry[8,17–19] (Fig. 1a).

A hallmark of both orbital and spin–orbital quadrupole current is the appearance of additional symmetry-breaking related to mirror, inversion or roto-inversion transformations, more generically known as chirality. A chiral electronic ordering may therefore be realized uniquely as a consequence of the intrinsic spin and orbital structure of the charge currents[20–22]. Chirality is known to set out several unconventional forms of transport and magnetism[2,23–28]. However, chiral effects have been difficult to detect because their electronic signature is weak. So far their measurements have been limited to only a few material-specific cases[1,29–31].

In a metallic state, chiral orders are imprinted on the spin and orbital textures of the electronic states close to the Fermi level[32–34]. The action of mirror- and time-reversal symmetries connects the amplitude of spin and orbital angular momentum of the electron states at symmetry-related momenta. For instance, for time-reversal symmetric electronic states (Fig. 1b), the associated spin angular momentum at symmetry related crystal wavevectors, $+\mathbf{k}$ and $-\mathbf{k}$, must have opposite orientations, that is, $(-\mathbf{k}, \uparrow)$ transforms into $(+\mathbf{k}, \downarrow)$[35]. The same behaviour applies to the orbital angular momentum $L$. However, for mirror-symmetric electronic states, owing to the axial nature of $S$ and $L$, after mirror transformation the components lying in the mirror plane change sign, whereas those perpendicular to the plane remain unchanged. Apart from the dipolar one, the interaction between spin and orbital degrees of freedom can set out physical observables with tensorial character. Here the time-reversal and mirror

[1]Department of Molecular Sciences and Nanosystems, Ca' Foscari University of Venice, Venice, Italy. [2]Istituto Officina dei Materiali, Consiglio Nazionale delle Ricerche, Trieste, Italy. [3]Institute of Theoretical Physics, Jagiellonian University, Kraków, Poland. [4]International Centre for Interfacing Magnetism and Superconductivity with Topological Matter, Institute of Physics, Polish Academy of Sciences, Warsaw, Poland. [5]Dipartimento di Fisica "E. R. Caianiello", Università di Salerno, Fisciano, Italy. [6]Istituto SPIN, Consiglio Nazionale delle Ricerche, Fisciano, Italy. [7]Synchrotron SOLEIL, Saint-Aubin, France. [8]Department of Physics and Astronomy, Interdisciplinary Nanoscience Center, Aarhus University, Aarhus, Denmark. [9]Dipartimento di Fisica, Politecnico di Milano, Milan, Italy. [10]Dipartimento di Fisica, Università degli Studi di Milano, Milan, Italy. [11]Department of Physics and Astronomy, Seoul National University, Seoul, Korea. [12]Istituto SPIN, Consiglio Nazionale delle Ricerche, Naples, Italy. [13]These authors contributed equally: Federico Mazzola, Wojciech Brzezicki. ✉e-mail: federico.mazzola@unive.it; mario.cuoco@spin.cnr.it; antonio.vecchione@spin.cnr.it

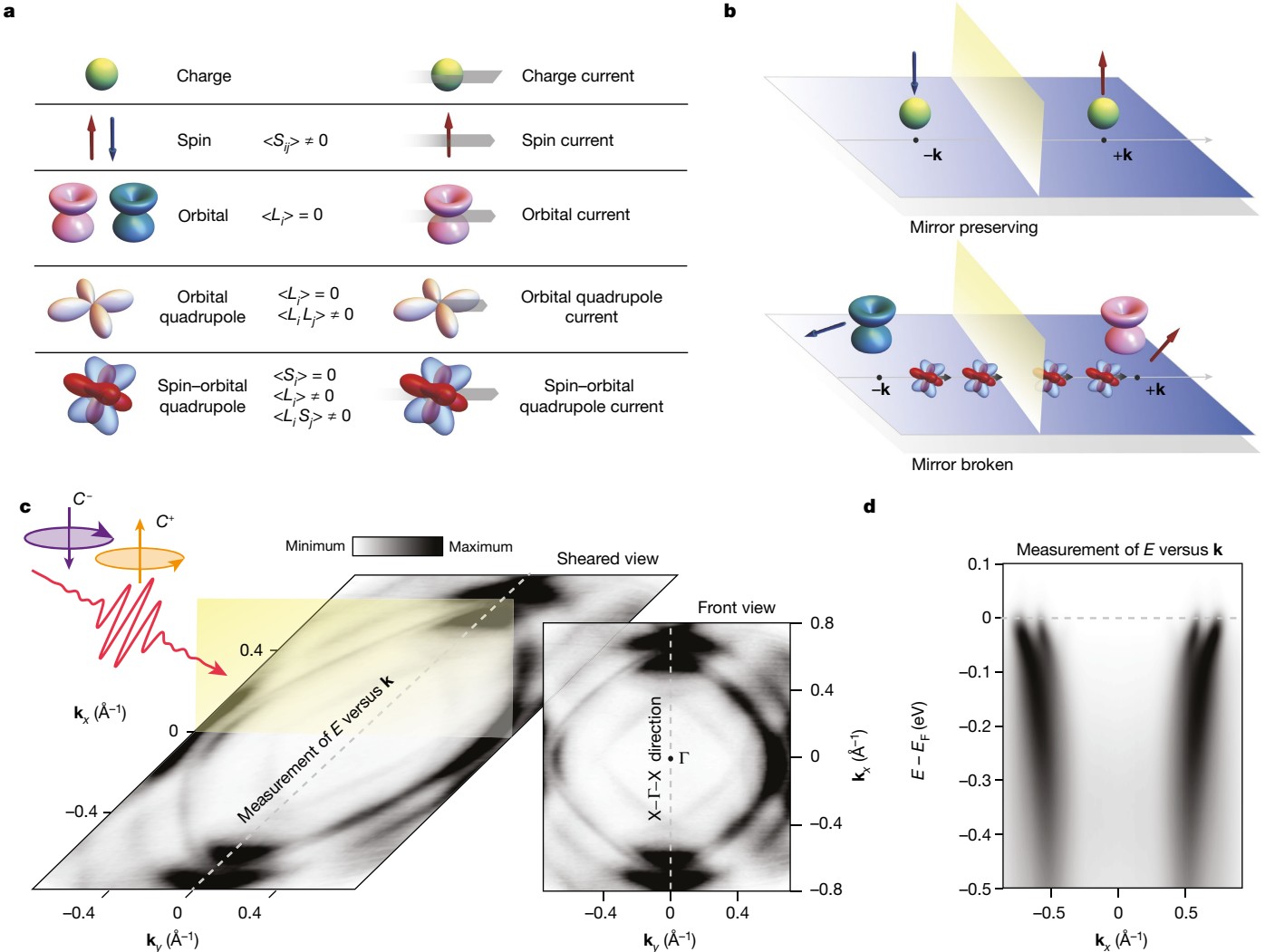

**Fig. 1 | Currents and symmetries in an electronic system. a**, The possible charge, spin and orbital currents that can be created in a material. The charge can give rise through its motion to a conventional current (the charge current), but spin and orbital dipoles or quadrupoles can also generate more-complex types of current. Indeed, the spin ($S$) and orbital ($L$) angular momentum are pseudovectors that change sign where there is time-reversal symmetry, and then the spin and orbital currents carrying spin or orbital dipoles are time-reversal conserving. Instead, the currents carrying orbital or spin–orbital quadrupoles break time-reversal symmetry and yield non-vanishing amplitudes for the dipole and quadrupole observables at a given momentum. **b**, Examples of mirror-preserving (top) and mirror-broken (bottom) configurations.

In a system that preserves time reversal, a charge with its spin at a certain positive momentum, under the action of such symmetries, goes into a charge with opposite spin (directed in the same direction but opposite in sign) at negative (symmetry related) momentum. For mirror-symmetric configurations, the sign change occurs when the spin lies in the mirror plane, as shown here. Instead, with currents included, strong asymmetry in their product $LS$ occurs. **c**, This experimental configuration with circularly polarized light ($C^{+,-}$) was used to measure the asymmetry of $LS$ caused by the chiral current-driven breaking of mirror symmetry. The Fermi surface of $Sr_2RuO_4$ is used here as a test bed for our theory. **d**, The binding energy ($E - E_F$) of the electrons is shown as a function of momentum ($\mathbf{k}$) for $Sr_2RuO_4$.

symmetries are also expected to affect the behaviour of the spin–orbital ($L_i S_{m\mathbf{k}}$) and orbital ($L_i L_{m\mathbf{k}}$) quadrupole components $\{i, m\} = x, y, z$ when probed at symmetry-related momenta. This means that the spin–orbital dipolar and quadrupolar structures are the relevant observables of the onset of symmetry breaking and can be used to assess the nature of the realized electronic ordering. For a ground state hosting chiral currents, there is a lack of symmetry in the combination of $L$ and $S$. In this situation, the spin–orbital texture of the electronic states at the Fermi level exhibits a distinctive behaviour: spin–orbital chiral currents give rise to orbital moments with the same parity as for mirror-symmetric systems, although spin–orbital quadrupoles have neither a time-symmetric nor a mirror-symmetric profile. It is this physical case that we explore in this work.

To exemplify the concept of a chiral metal, we can make an analogy with chiral crystals and their symmetry properties. We can then generally identify a chiral metal with an electronic state that has a well defined handedness, owing to the lack of inversion, mirror or other roto-inversion symmetries[36,37]. In this study, we start from this description to introduce the concept of a surface spin–orbital chiral metal to indicate a conducting electronic state of matter that has a well defined handedness, owing to an interaction driven by a magnetochiral order that lacks mirror symmetries, resulting from the internal spin–orbital structure, but has the same translational symmetry as the hosting crystal.

Here we show the relationship between the spin–orbital textures of the electronic states, for both dipolar and quadrupolar channels, and the consequential occurrence of a chiral electronic ordering. By supporting the theory with circularly polarized, spin-selective, angular-resolved photoelectron spectroscopy, we introduce a methodology to probe otherwise undetectable symmetry-broken chiral

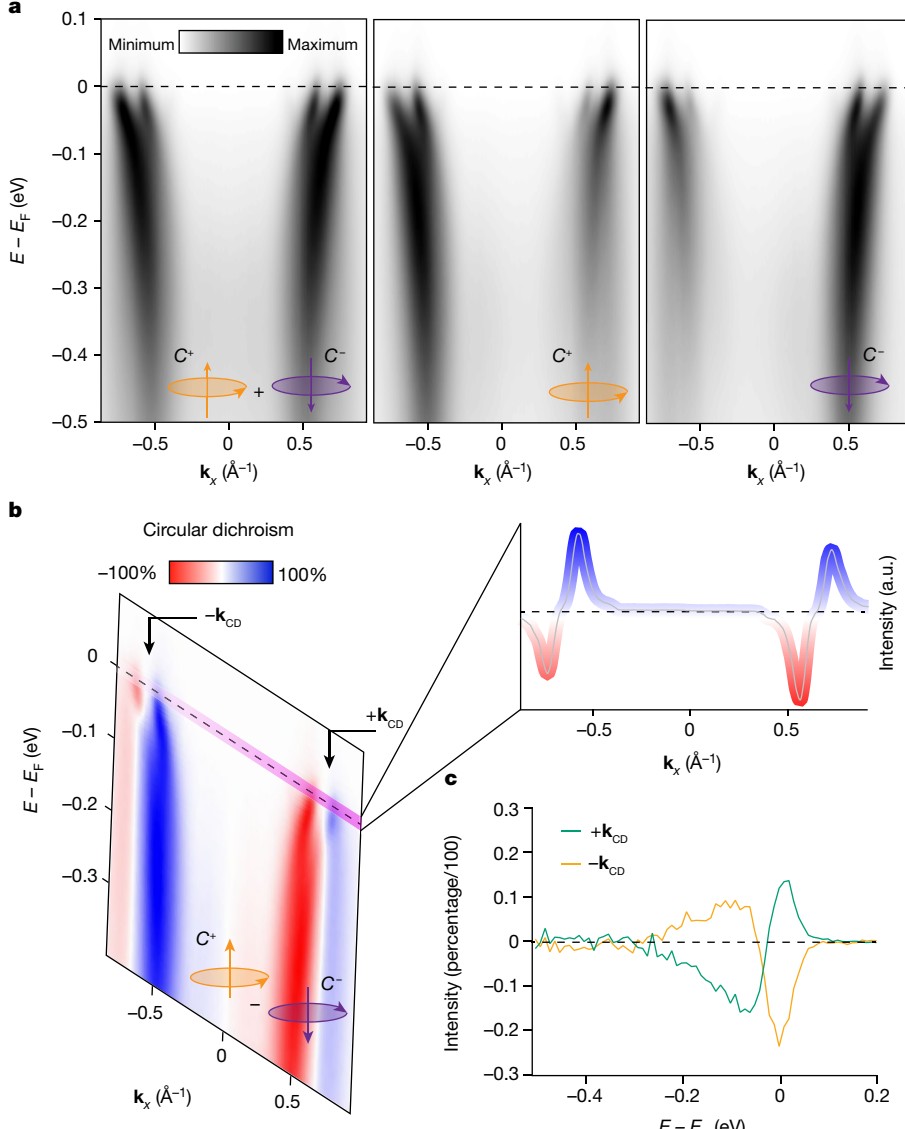

**Fig. 2 | CP-spin-integrated ARPES. a**, Left: unpolarized ARPES spectrum from $Sr_2RuO_4$ along the direction orthogonal to the crystal mirror plane, corresponding to the dashed line in Fig. 1c. Middle and right: the spectrum has been obtained by summing both contributions from right- (middle) and left-circularly (right) polarized light. Here we refer to this as $C^{+,-}(\pm k, \uparrow, \downarrow)$ to indicate signals from right- or left-circularly polarized light, collected at momentum $\pm k$, and with a spin-up or spin-down channels, respectively.

**b**, Circular dichroism of ARPES spectrum obtained by subtracting the contributions from right- and left-circularly polarized light. Remarkably, the signal changes sign from $+k$ to $-k$, with incoming light within the mirror plane. The asymmetry seen is discussed in both the main text and Methods. **c**, Energy-dependent circular dichroism collected with spin-detector (VLEED) at the **k**-points indicated in **b**.

electronic states. To do this, we use the archetypal quantum material $Sr_2RuO_4$ (see Methods for details of growth and measurements) and reveal signatures of a broken symmetry phase[8], compatible with the formation of spin–orbital quadrupole currents at the surface of the material.

## Dichroic and spin–dichroic photoemission effects

As anticipated, assessing whether the profile of the $LS$ quadrupole components has a mirror- and time-broken character is key to the detection of electronic phases with electronic chiral currents. $Sr_2RuO_4$ is an ideal candidate to host symmetry-broken chiral ground states because of its low-energy muon spin-spectroscopy[8] and scanning-tunnelling microscopy measurements[38], which reveal the existence of unconventional magnetism and electronic ordering forming at the surface.

Despite this, there have so far been no accepted methodologies to ascertain the existence of such symmetry-broken chiral states. According to our theory, to detect the action of symmetries on $LS$, we have to study the out-of-plane components of the orbital angular momentum ($L_z$) and spin ($S_z$). Experimentally, the $LS$ asymmetry can be studied by circularly polarized, spin-selective, angular-resolved photoelectron spectroscopy[35,39] (CP-spin-ARPES). This approach requires extreme caution in the alignment and geometry of the apparatus (Fig. 1c). Indeed, photoelectrons from circularly polarized light host a combination of intrinsic and geometric matrix elements[40,41]. These can, nevertheless, be disentangled[41] (Methods).

In Fig. 2a, we show CP-ARPES spectra, collected in the geometry of Fig. 1c, that are compatible with signals coming from both the bulk and surface states. The latter appear weaker in spectral intensity but are clearly visible after increasing the contrast (Methods).

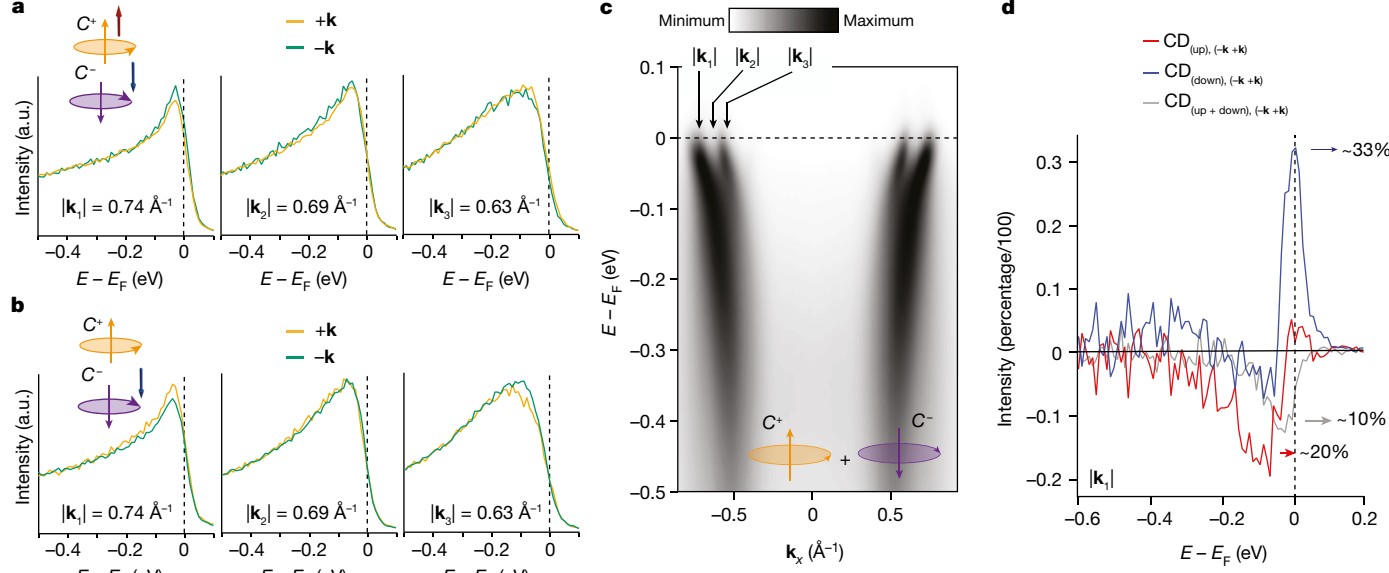

**Fig. 3 | CP-spin-resolved ARPES. a**, EDCs taken at six selected momenta ($\pm\mathbf{k}_i$, where $i = 1, 2$ or $3$) with fixed spins and circular polarizations. In particular, the orange curves are obtained by measuring the EDCs at positive $\mathbf{k}$ values, right-circularly polarized light and spin-up channel ($C^+(\mathbf{k}, \uparrow)$), whereas the green curves are obtained with negative $\mathbf{k}$ values, left-circularly polarized light and spin-down channel ($C^+(-\mathbf{k}, \downarrow)$). **b**, ARPES spectra with reversed spin and circularly polarized light configurations. The orange curves refer to $C^+(-\mathbf{k}, \uparrow)$, whereas the green curves are obtained for $C^-(\mathbf{k}, \downarrow)$. **c**, ARPES image indicating the $\mathbf{k}$ values at which the EDCs have been taken. It is noted that the configurations in **a** and **b** show a difference that is larger than the experimental uncertainty. **d**, The amplitudes of the circular dichroism (at $\mathbf{k}$ summed up to see the actual

residual) are reported for both spin-integrated and spin-resolved measurements. The data show that the spin-integrated signal (grey curve) shows a finite value as large as 10% (which is also similar to the experimental uncertainty of 8%, as shown in ref. 39), but the spin-resolved channels show a notably larger amplitude, by a factor of 2 and 3 for up and down channels, respectively. The amplitude values have been extracted from the data shown in **a** and **b** and in Extended Data Fig. 3, after including the Sherman function and calculating the true spin polarization, as described in Methods. The other indicated $\mathbf{k}$ points, as well as the dichroic amplitude in terms of the momentum distribution curve, are shown in Extended Data Figs. 4 and 5, and corroborate the validity of our result.

From left to right, Fig. 2a shows spectra with unpolarized, right- and left-circularly polarized photons. The CP-spectrum shows an overall symmetric intensity pattern between features at $+\mathbf{k}$ and at $-\mathbf{k}$. From these spectra, the circular dichroism (CD) is extracted (Fig. 2b) and the signal CD($+\mathbf{k}$) becomes $-$CD($-\mathbf{k}$) at opposite momenta, consistent with previous studies[42]. This behaviour is also seen in the momentum distribution curve at the Fermi level (Fig. 2b, inset). Importantly, studying the spin-integrated circular dichroism in Fig. 2b reveals a small asymmetry in the residual of the amplitudes (Extended Data Figs. 5 and 6) that can be as large as 10%. This value is slightly bigger than the estimated experimental error on the dichroism for this experimental set-up (around 8%). Furthermore, there might be a component of asymmetry related to the character of the chiral electronic ordering (discussed in Supplementary Information). However, such an asymmetry remains notably smaller than the one measured for spin-resolved signals. For completeness, spin-integrated data are collected with a (VLEED)-type spin detector (Fig. 2c at two $\pm\mathbf{k}$ points) and show the same behaviour.

Importantly, the spin-integrated data in Fig. 2c are not only consistent with other ARPES studies, but also they do not reveal mirror-symmetry breaking relatable to anomalous behaviour of $L_z$ ($z$ being the direction perpendicular to the surface). However, quantities such as spin–orbital chiral currents, which depend on the spin as much as on the orbital angular momentum, cannot be imaged by standard circularly polarized ARPES. Here we refer to $C^{+,-}(\pm\mathbf{k}, \uparrow, \downarrow)$ to indicate signals from right- (or left-) circularly polarized light, collected at momentum $\pm\mathbf{k}$, and with a spin-up (or spin-down) components, respectively. In a perfectly symmetry-preserving situation, the circularly polarized spin-ARPES intensity transforms under the mirror operator from $C^+(+\mathbf{k}, \uparrow)$ to $C^-(-\mathbf{k}, \downarrow)$ (or equivalently from $C^+(-\mathbf{k}, \uparrow)$ to $C^-(+\mathbf{k}, \downarrow)$). This means that $C^+(+\mathbf{k}, \uparrow)$ and $C^-(-\mathbf{k}, \downarrow)$ (or $C^+(-\mathbf{k}, \uparrow)$ to $C^-(+\mathbf{k}, \downarrow)$) are expected to be the

same under mirror symmetry in the case that the latter is preserved[35]. However, if chiral currents are present along the surface, the mirror symmetry is broken and these quantities are no longer equivalent. We tested this scenario with CP-spin-ARPES and the results are shown in Fig. 3a,b: we observe subtle differences at different $\mathbf{k}$ points, as noted in Fig. 3c, between $C^+(+\mathbf{k}, \uparrow)$ and $C^-(-\mathbf{k}, \downarrow)$ (or $C^+(-\mathbf{k}, \uparrow)$ and $C^-(+\mathbf{k}, \downarrow)$). These differences, despite being small, result instead in a sizeable asymmetry in the amplitudes of spin-up and spin-down dichroism (Fig. 3d). Note that for the latter, positive and negative momenta have been summed, compensating for possible instrumental asymmetry of the measurements.

Such a mirror-symmetry breaking in the amplitude of the spin dichroism seems to be compatible with the presence of spin–orbital quadrupole currents, as predicted by the theory. Furthermore, the finite dichroism difference observed experimentally is not seen for a temperature of 77 K, higher than the magnetic transition temperature as indicated by muon spectroscopy[8], but this requires further investigation because the thermal broadening becomes substantially more pronounced (Methods). The important finding here is that the estimated difference in spin from the dichroic signals is up to three times larger than the spin-integrated one. In Fig. 3d, the amplitude of the spin-integrated dichroic signal is approximately 10% (grey curve), whereas the spin-resolved signal is as high as 20% for spin-up (red curve) and 30% for spin-down (blue curve); see Supplementary Information for a possible explanation of the spin-integrated asymmetry observed. We emphasize that a quantitative analysis is difficult and the signals detected are subtle. To quantify these effects correctly, future measurements as a function of photon energy and various geometries will be desirable. Nevertheless, the presence of a sizeable asymmetry in the amplitude signal for $\mathbf{k}$ and $-\mathbf{k}$ is observed and this is consistent with the theoretical predictions.

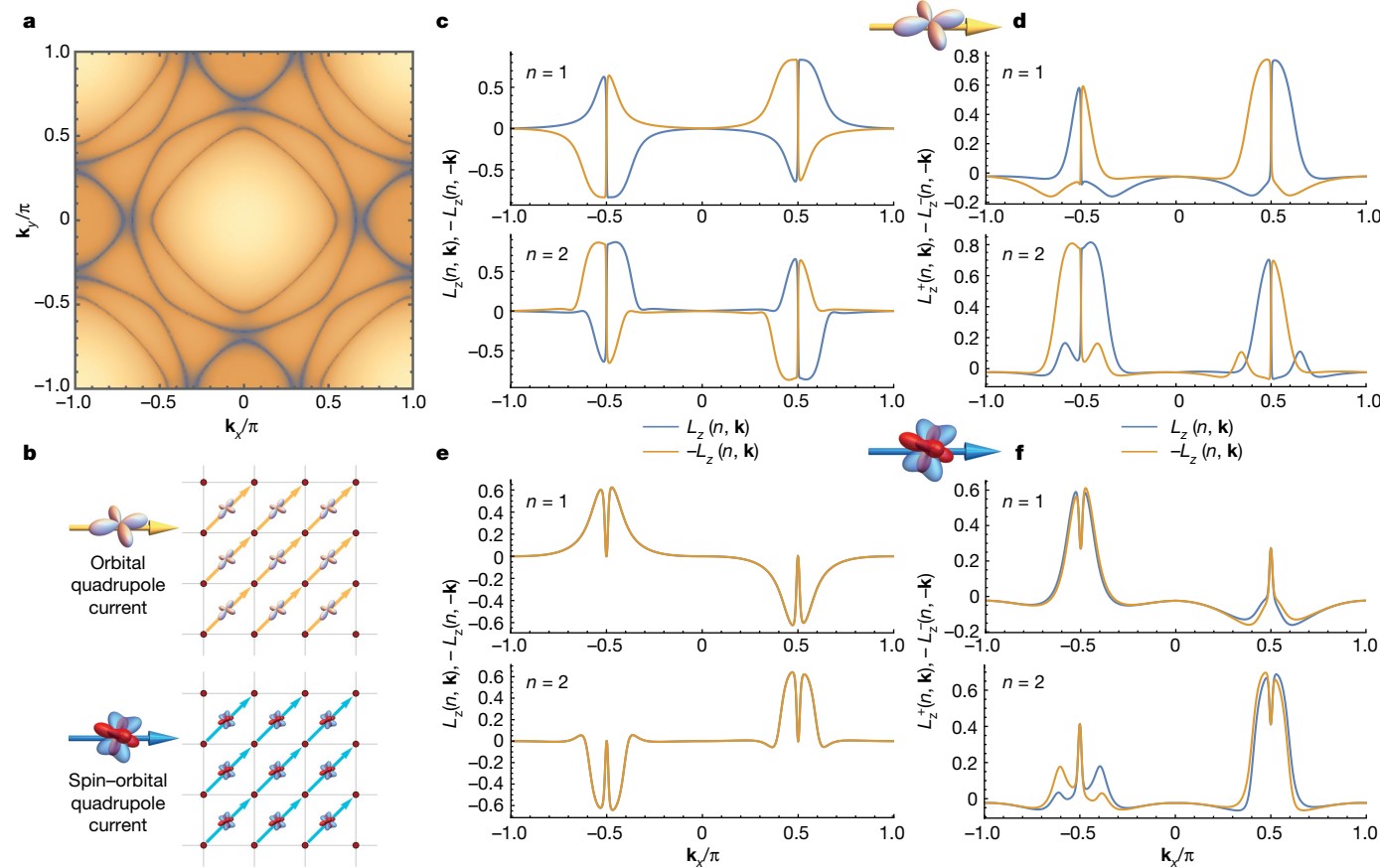

**Fig. 4 | Orbital and spin–orbital textures in the presence of chiral currents.** **a**, The computed Fermi surface of $Sr_2RuO_4$. **b**, A broken-symmetry state with an electronic pattern marked by either orbital-quadrupole (top) or spin–orbital quadrupole (bottom) currents. The sketch indicates a current in real space connecting the ruthenium sites along the [110] direction. **c**, Electronic phase with chiral orbital-quadrupole currents: amplitude of the orbital angular momentum $L_z(n, \mathbf{k})$ of the bands described by the eigenfunctions of the Hamiltonian $|\psi_{n,\mathbf{k}}\rangle$ with $n = 1, 2$ evaluated along the $\Gamma$–X direction $(L_z(n, \mathbf{k}) = \langle\psi_{n,\mathbf{k}}|\hat{L}_z|\psi_{n,\mathbf{k}}\rangle)$. For clarity, we plot both $L_z(n, \mathbf{k})$ (blue) and $-L_z(n, -\mathbf{k})$ (orange) for any given momentum $\mathbf{k}$ to directly compare the amplitudes at opposite momenta. **d**, Electronic phase with chiral orbital-quadrupole currents: amplitude of the spin-projected orbital angular momentum related to the out-of-plane spin-up (+) and spin-down (−) components, as selected by the projector $(1\pm\hat{s}_z)$. The amplitude is given by $L_z^{\pm}(n, \mathbf{k}) = \langle\psi_{n,\mathbf{k}}|(1\pm\hat{s}_z)\hat{L}_z|\psi_{n,\mathbf{k}}\rangle$. The amplitudes of $L_z(n, \mathbf{k})$ and $L_z^{\pm}(n, \mathbf{k})$, shown in **c** and **d**, do not show any symmetry and do not match at $\mathbf{k}$ and $-\mathbf{k}$. **e**, Electronic phase with chiral spin–orbital quadrupole currents with antisymmetric $L$ and $S$ content with respect to the current flow direction $\mathbf{k}$, that is, $\mathbf{k} \cdot (\hat{\mathbf{L}} \times \hat{\mathbf{s}})$; for this configuration, $L_z(n, \mathbf{k})$ and $L_z(n, -\mathbf{k})$ coincide. **f**, Electronic phase with chiral spin–orbital quadrupole currents with antisymmetric $L$ and $S$ combination: spin-projected orbital moment $L_z^{\pm}(n, \mathbf{k})$ at opposite momenta are unequal in amplitude. Similar trends occur for the other bands (Supplementary Information).

## Chiral current phase

We now analyse our experimental results from a theoretical point of view. This can be done with a two-dimensional tight-binding description of the electronic structure of $Sr_2RuO_4$ on the basis of the ruthenium $d$ orbitals ($d_{xy}$, $d_{xz}$ and $d_{yz}$) in such a way as to capture the profile of the experimental Fermi surface (Fig. 4a). In this model, we include the broken-symmetry states hosting orbital and spin–orbital quadrupole currents that are driven by the $d$–$d$ Coulomb interactions (Supplementary Information). The internal structure of the charge currents is provided by either the orbital quadrupole $\hat{L}_p\hat{L}_q$ or by the spin–orbital quadrupole $\hat{L}_p\hat{s}_q$ tensors, with $\{p, q\} = x, y, z$ (see Fig. 1a for differences between these currents). For a given direction $k_l$ in momentum space, the amplitude of the charge current propagating through the lattice can in principle contain various components of the type $\hat{j}_o^l = \sin(k_l)\hat{L}_p\hat{L}_q$ and $\hat{j}_{so}^l = \sin(k_l)\hat{L}_p\hat{s}_q$ for the orbital and spin–orbital quadrupole, respectively (note that the first does not contain any spin channel, whereas the second one does). In the presence of currents flowing along for example $l = x$ that break all mirror symmetries (that is, $M_{j=x,y,z}$), the orbital and spin–orbital quadrupoles have to include components with $\hat{s}$ and $\hat{L}$ that are perpendicular to $x$, and are therefore lying in the $y$–$z$ planes. For instance, a term of the type $\sin(k_x)\hat{L}_y\hat{L}_z$ does not preserve the mirror symmetry for any choice of the $M_{j=x,y,z}$ transformations. The latter is exactly the case for $Sr_2RuO_4$, in which a uniform charge current flows with the momentum aligned along the $l = [110]$ direction of the ruthenium lattice (Fig. 4b). This pattern is compatible with symmetry-allowed loop currents involving charge current flowing from ruthenium to oxygen atoms at the octahedra length scale[8]. The qualitative outcomes of the results are not altered by surface reconstruction or by having currents flowing along other symmetry directions (Supplementary Information). We select a spin–orbital chiral state that breaks the $C_4$ rotational symmetry because it is compatible with the findings from scanning tunnelling microscopy[38].

However, spin–orbital chiral states with loop currents that are rotational invariant can be also constructed. They break translational symmetry and do not modify the qualitative outcomes of the analysis (Supplementary Information).

Furthermore, the charge current along the [110] axis is directly relevant when probing the electronic states along the corresponding direction of the Brillouin zone (along $\Gamma$–X in our experiment). To evaluate the orbital and spin–orbital textures for the electronic states

in the presence of either orbital or spin–orbital quadrupole currents, we focus on such directions. For each band eigenstate $|\psi_{n,\mathbf{k}}\rangle$ at a given momentum $\mathbf{k}$, we determine the amplitude of the out-of-plane orbital moment $L_z(n,\mathbf{k}) = \langle\psi_{n,\mathbf{k}}|\hat{L}_z|\psi_{n,\mathbf{k}}\rangle$ and the spin-projected orbital moment $L_z^{\pm}(n,\mathbf{k}) = \langle\psi_{n,\mathbf{k}}|(1\pm\hat{s}_z)\hat{L}_z|\psi_{n,\mathbf{k}}\rangle$ (that is, the out-of-plane spin-up (+) and spin-down (−) components, as singled out by the projector $(1\pm\hat{s}_z)$). These observables are related to the dichroic and spin-dichroic amplitudes probed by ARPES, respectively (Supplementary Information). For clarity and simplicity, in Fig. 4c–f we show only two representative bands, for example $n = 1, 2$ (the behaviour for the other bands is qualitatively similar, as shown in Supplementary Information). As shown in Fig. 4c, a broken-symmetry state with orbital quadrupole currents exhibits an asymmetry in the orbital angular moment at opposite momenta: $L_z(n,\mathbf{k}) \neq -L_z(n,-\mathbf{k})$. Moreover, both the amplitude and the sign of $L_z^{\pm}(n,\mathbf{k})$ and $L_z^{\pm}(n,-\mathbf{k})$ are dissimilar (Fig. 4d). Instead, for spin–orbital chiral currents $j_{so}^l$ (Fig. 4b) with an antisymmetric combination of $L$ and $S$, that is, $j_{so}^l = \sin(\mathbf{k}_l)(\hat{L}\times\hat{s})_l$, we find that the orbital angular momentum turns out to be antisymmetric at $\mathbf{k}$ and $-\mathbf{k}$, namely, $L_z(n,\mathbf{k}) = -L_z(n,-\mathbf{k})$ (Fig. 4e), whereas the spin-projected orbital moment does not exhibit any symmetry relation among the states at the opposite momentum (Fig. 4f). Importantly, the last asymmetry is the same as the one observed experimentally by CP-spin-ARPES. One way to grasp the origin of this behaviour is to inspect the structure of the equations of motion for the amplitudes of the orbital $L_z(n,\mathbf{k})$ and spin $s_z(n,\mathbf{k})$ moments (Supplementary Information). The chiral currents lead to spin–orbital torques that in the case of the spin–orbital quadrupole current result in a balance such as the amplitude of the orbital moment having symmetric behaviour at symmetry-related momenta.

We also addressed the role of sublattice chiral currents caused by the surface reconstruction resulting from the octahedral rotations (Supplementary Information). When we consider a non-homogeneous state with a staggered amplitude modulation of the currents, we find that a small asymmetry of the orbital moment is obtained. Nevertheless, its amplitude is substantially smaller than that of the spin-projected orbital moment (Supplementary Information). This implies that non-homogeneous chiral currents are also compatible with the observation of a much larger spin-dichroic asymmetry relative to the dichroic one. This behaviour holds independently of the selected band and momentum, and differently from that of magnetic states, such as antiferromagnetic ones, with a pattern in the spin and orbital moments that break time and mirror symmetries (Supplementary Information).

## Conclusions

We developed a spin–orbital- and angular-momentum-sensitive methodology based on CP-spin-ARPES that is able to investigate symmetry breaking and is compatible with the existence of spin–orbital chiral currents. We used it to study $Sr_2RuO_4$ but it applies to all chiral surface metals and constitutes a tantalizing experimental way to detect symmetry-broken chiral states. However, because the effect is subtle, we cannot exclude the possibility that the observed symmetry breaking might arise from other real-space chiral orderings that break both time and mirror symmetries. Nevertheless, our work stimulates the combined use of circular dichroism and spin-selective photoemission to investigate how the three quantities $L$, $S$ and $LS$ behave and reveal their relationship to crystal symmetries, which are markers of hidden ordered phases.

The spin-dichroic signal we used to detect the putative presence of spin–orbital quadrupole currents at the surface of $Sr_2RuO_4$ can be used without restrictions in other quantum materials, even when currents appear in the bulk of a (centrosymmetric) crystal. In this situation, the currents might preserve the combination of time reversal with inversion symmetry, with the consequent absence of dichroism. Nevertheless,

the asymmetry of the spin-dichroic signal could still be visible and therefore represents an efficient diagnostic tool for spin–orbital chiral metallic phases.

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

## Methods

The samples of $Sr_2RuO_4$ were grown using the floating-zone technique, following a previously published procedure[43]. Single crystals were postcleaved in an ultrahigh vacuum at a base pressure of $1 \times 10^{-10}$ mbar and a temperature of 20 K (and 77 K). The temperature was kept constant throughout the measurements. The experiment was performed at the NFFA–APE Low Energy beamline laboratory at the Elettra synchrotron radiation facility and designed with an APPLE-II aperiodic source for polarized extreme UV radiation and a vectorial twin-VLEED spin-polarization detector downstream of a DA30 Scienta ARPES analyser[44]. The photon energy used for our measurements was 40 eV, which was found to maximize the spectral intensity, as shown previously[45]. The energy and momentum resolutions were better than 12 meV and 0.018 Å$^{-1}$, respectively. Importantly, as already mentioned, to eliminate the geometrical contribution to the circular polarization, the crystals were aligned as in Fig. 1c,d. For completeness, seminal works on ARPES and dichroism that might aid the understanding of our measurements can be found in refs. 39,41,46–48.

In the following sections, we report additional measurements that help to corroborate the message and conclusions given in the main text.

### Sample alignment and experimental geometry

When using circularly polarized light, the disentanglement between geometrical and intrinsic matrix elements is crucial but problematic. A solution is to have the incoming radiation exactly within one of the mirror planes of the system studied and to measure in the direction orthogonal to that plane, as we show in Fig. 1c. In such a configuration, the differences in the CP-spin-ARPES signal can be attributed to intrinsic differences in $LS$, and the geometrical contributions are well defined. In this regard, it is of paramount importance to align the sample carefully. In the present case, the symmetric character of the material's Fermi surface[45,49,50] allows us to carefully align the sample with the incoming beam of photons lying in a mirror plane. The alignment of the sample was carried out by monitoring the experimental Fermi surface and by making sure that the analyser slit direction was perpendicular to the mirror plane. As shown in Extended Data Figs. 1 and 2, we estimated our alignment to be better than 0.9° from the ideal configuration, a value within the uncertainty considering our angular azimuthal precision (about 1°). Furthermore, different samples gave us the same results, corroborating the robustness of the measurement outputs within this azimuth uncertainty.

In the NFFA–APE Low Energy beamline laboratory, our sample was placed in the manipulator in normal emission conditions, with the synchrotron light impinging on the sample surface at an angle of 45°. This means that standard linear polarizations, such as linear vertical and linear horizontal (Extended Data Fig. 1), would act differently on the matrix elements' selection rules. In particular, linear vertical light would be fully within the sample plane, whereas linear horizontal light would have one component within the plane and one out of plane (with 50% intensity each). Now, when using circularly polarized light, to distinguish between real and geometrical matrix element effects, the incoming light needed to be aligned within the experimental error, within one of the mirror planes of the sample.

To estimate the azimuthal value we fitted the **k**-loci of the Fermi surface contours (red markers in Extended Data Fig. 2a,b) and we then aligned the horizontal and vertical axes (see 'Details of the fitting'). In our configuration, there is negligible misalignment between the states at positive and negative values of **k** (Extended Data Fig. 2c,d). In Extended Data Fig. 2, we show that by extracting momentum distribution curves (coloured horizontal lines in Extended Data Fig. 2c), the peak positions are symmetric within the resolution of the instrument (12 meV for energy and 0.018 Å$^{-1}$). We can therefore confidently perform the measurements shown in the main text.

### Details of the fitting

The **k**-loci of the Fermi surfaces shown in Extended Data Fig. 2a,b and the positions of the peaks in Extended Data Fig. 1d have been extracted by fitting the ARPES data. The fitting procedure used is standard and consists of fitting both energy distribution curves (EDCs) and momentum distribution curves by using Lorentzian curves convoluted by a Gaussian contribution that accounts for the experimental resolutions. Then, as part of the fit results, we extracted the **k** positions of the peaks, which are shown as red markers in Extended Data Fig. 2 and the values in Extended Data Fig. 2d.

### Spin-ARPES data

To obtain the values reported, the spin data shown have also been normalized to include the action of the Sherman function of the instrument. In particular, the data for spin-up and spin-down channels have been normalized to their background, so they matched in both cases. In the present study, the background normalization was done on the high-energy tails of the EDCs far from the region where the spin polarization was observed. After normalization, to extract the spin intensity, we used the following relations:

$$I^{TRUE}(\mathbf{k}, \uparrow) = \frac{I^{TOT}(\mathbf{k})}{2} \times (1 + P),$$

$$I^{TRUE}(\mathbf{k}, \downarrow) = \frac{I^{TOT}(\mathbf{k})}{2} \times (1 - P),$$

where $P$ is the polarization of the system, $I^{TRUE}$ is the intensity value (for either spin-up or -down species) obtained after inclusion of the Sherman (see below) function of the spin detector, and $I^{TOT} = I^{bg.norm}(\mathbf{k}, \uparrow) + I^{bg.norm}(\mathbf{k}, \downarrow)$ is simply the sum of the intensity for EDCs with spin-up and spin-down after normalization to the background. For the polarization $P$, the Sherman function from the instrument was included and defined as $\eta = 0.3$ (ref. 44). The Sherman function was calibrated from measurements on a single gold crystal. Therefore, $P$ is described by:

$$P(\mathbf{k}) = \frac{1}{\eta} \times \frac{I^{bg.norm}(\mathbf{k}, \uparrow) - I^{bg.norm}(\mathbf{k}, \downarrow)}{I^{bg.norm}(\mathbf{k}, \uparrow) + I^{bg.norm}(\mathbf{k}, \downarrow)}.$$

This procedure was done for all light polarizations. We also characterized the spin channels by using different polarization-vector directions, as shown in Extended Data Fig. 3.

### Dichroism and spin-dichroism amplitudes

A way to visualize the breaking of the time-reversal symmetry is to analyse the dichroic signal shown in Fig. 2c but resolved in the two different spin channels, up and down, which gives rise to different amplitude values when measured at ±**k** (expected for time-reversal symmetry breaking but not expected otherwise). We show this here at selected momentum values. The amplitude values have been extracted from the data shown in Fig. 3a and Extended Data Fig. 3, after including the Sherman function normalization.

To corroborate the claim in the main text, that is, the observation of a signal compatible with the existence of chiral currents, Extended Data Fig. 4 shows the relative amplitudes of the dichroic versus spin-dichroic signal. First, let us consider the spin-integrated dichroism shown in Extended Data Fig. 4a. Here, the orange and green curves represent positive and negative **k** values, respectively, and their behaviour is overall symmetric with respect to zero. However, a small asymmetry can still be noticed, estimated to be as large as 10%, which is close to a previously reported value[39] of 8%. As we will clarify from a theoretical point of view, a small degree of asymmetry in the spin-integrated dichroism can still be expected, although the amplitudes of the dichroism selected

in their spin channels are supposed to be larger. To demonstrate this difference, we have shown how the dichroism curves, resolved in their spin channels, up (red) and down (blue), appear at negative **k** (Extended Data Fig. 4d–f) and at positive **k** (Extended Data Fig. 4g–i). By also considering their residuals, we can compare them to the amplitude of the spin-integrated signal. We reported this comparison in Extended Data Fig. 5. The spin-down channel shows an amplitude as high as 30% and the spin-up one is as high as 20%. These values are three times and two times bigger, respectively, than the residual extracted for the spin-integrated signal. Such a large difference corroborates the validity of our methodology and the claims of our work. Note that summing the positive and negative momentum is also counteracting any possible effects caused by small sample misalignment.

## Data and temperature

For completeness, we also performed $C^+(+\mathbf{k}, \uparrow)$ and $C^-(-\mathbf{k}, \downarrow)$ on the sample after cleaving it, also at high temperature (70 K), which is above the magnetic transition of $Sr_2RuO_4$. We report the results in Extended Data Fig. 6. In particular, in Extended Data Fig. 6a–c, the top panels with blue lines show the difference between $C^+(+\mathbf{k}, \uparrow)$ and $C^-(-\mathbf{k}, \downarrow)$, normalized by their sum, at three values of **k** and at low temperature, but the bottom line is the same for the data collected at 70 K. If in the low-temperature configuration we observe a varying finite signal, at high temperature we did not see such a variation. It is important to mention that even with our resolution, we do not see any finite signal, although there might be some differences that could be observed above the magnetic transition, because it is likely that not all magnetic excitations are turned off immediately, although a reduction should be still observed. Furthermore, the high-temperature data are more noisy. Even if we cleaved the samples at high temperature, and the ARPES shown in Extended Data Fig. 6d,e confirms their presence, they are much weaker than at low temperature and are broadened thermally. Such a thermal broadening is not surprising to see in ARPES. Nevertheless, even with reduced intensity, the surface states are still clearly visible.

## Calibrating the VLEED

Within the uncertainty of the instrument (1° integration region), the VLEED has been calibrated by acquiring spin EDCs at various angles, both positive and negative, for the sample. This is done for both spin species and with the used light polarizations. In the present case, for consistency, we did this with circularly polarized light (both left- and right-handed). Afterwards, by summing both circular polarizations and both spin species, we can reconstruct the ARPES spectra (Extended Data Fig. 7). This procedure was done by using only the spin detector to directly access the probed states and be sure that, when selecting the angular values on the deflectors, we effectively probe the selected state.

## Uncertainties and additional calibration

To evaluate the uncertainty we used a controlled and known sample with no asymmetries in the dichroic signal, as in our previous work[39]. We used a kagome lattice because at the Γ point there is a well defined energy gap, opened by the action of spin–orbit coupling. Furthermore, at this point the bands are spin-degenerate; the system is also not magnetic. This allowed us to check the asymmetry, not only in the circular dichroism signal, but also in the spin-resolved circular dichroism. We estimated the uncertainty to be approximately 10% on the residual of

the dichroism. Note that this is also consistent with that obtained by standard ARPES in our set-up: at the centre of the Brillouin zone, the difference between circular right- and circular left-polarized spectra (each spectrum was normalized by its own maximum intensity beforehand) is indeed 10%.

## Data availability

The data that support the findings of this study are available at https://doi.org/10.5281/zenodo.10350799 (ref. 51).

## Code availability

The code that supports the findings of the study is available from the corresponding authors on reasonable request.

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

**Acknowledgements** F.M. acknowledges the SoE action of PNRR, SOE_0000068. M.C., R.F., M.T.M. and A.V. acknowledge support from the EU's Horizon 2020 Research and Innovation programme, grant agreement 964398 (SUPERGATE). M.C. and G.P. acknowledge support from PNRR MUR project PE0000023-NQSTI. M.C., R.F., A.G. and A.V. acknowledge partial support by the Italian Ministry of Foreign Affairs and International Cooperation, grant KR23GR06. R.F., A.G., A.V. and M.C. acknowledge support by the PRIN 2020 project Conquest funded by the Italian Ministry of University and Research (Prot. 2020JZ5N9M). W.B. acknowledges support from Narodowe Centrum Nauki (NCN, National Science Centre, Poland) project 2019/34/E/ST3/00404 and partial support by the Foundation for Polish Science through the IRA Programme co-financed by the EU within SG OP. This work was performed in the framework of the Nanoscience Foundry and Fine Analysis (NFFA-MUR Italy) facility. J.A.M. acknowledges support from DanScatt (7129-00011B).

**Author contributions** A.V., M.C. and F.M. devised the project with the help of all the authors. A.V., M.C. and F.M. conceived the ARPES experiment on the basis of theoretical models developed with the help of W.B., M.T.M. and C.O. W.B. performed the analysis of the orbital and spin–orbital textures for the chiral currents phases. The samples were grown by R.F. and A.V. and characterized by A.G., R.F. and A.V. F.M. and M.C. wrote the paper, with input from all authors. A.V., M.C., F.M., W.B., M.T.M., A.G., C.B., J.A.M., D.D.F., A.C., J.F., G.R., P.O., S.K.C., S.P.C., G.P., A.J., V.P., I.V., C.K., F.M.-G., R.F. and C.O. discussed the results and their interpretation and revised the paper.

**Competing interests** The authors declare no competing interests.

**Additional information**
**Correspondence and requests for materials** should be addressed to Federico Mazzola, Mario Cuoco or Antonio Vecchione.

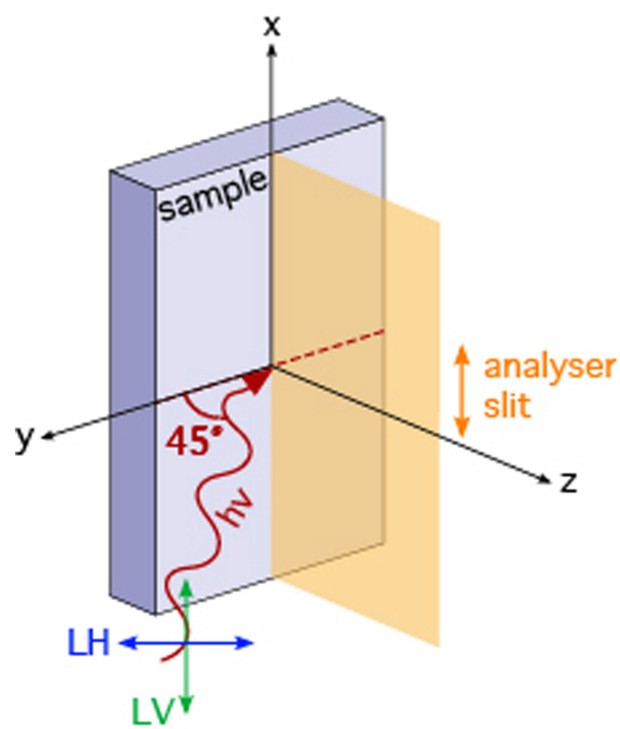

**Extended Data Fig. 1 | Photoemission experimental geometry.** The sample, represented by the purple box, is such that the incoming synchrotron radiation (red wavy arrow) impinges with an angle of 45° with respect to its surface. In this configuration, with linear polarizations, we would have linear vertical (LV, green double-headed arrow) lying completely on the sample surface. Instead, linear horizontal (LH, blue double-headed arrow) would have both in- and out-of-plane components, projected along the $y$- and $z$-axis, respectively. The slit of the analyser is along the scattering plane (vertical slit).

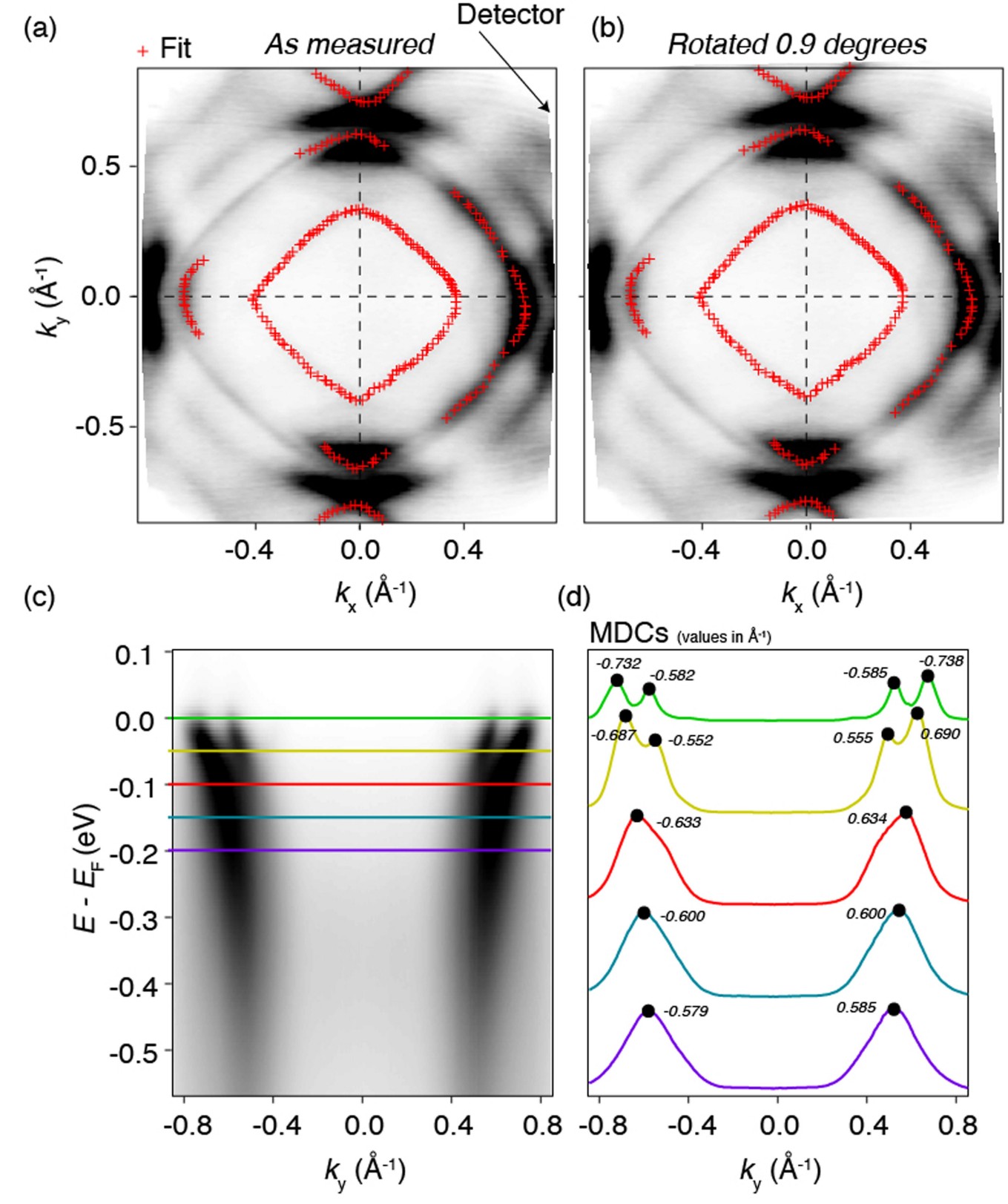

**Extended Data Fig. 2 | ARPES identification of the surface states and sample alignment. a** Fermi surface collected at 40 eV (sum of the two circularly polarized lights) showing both bulk bands and surface states. The latter are weaker than the bulk in intensity but still visible. To better appreciate the precise sample alignment we fitted the data and extracted the *k* positions, reported in the image as red markers. The mirror plane deviates from the (**b**) ideal condition by 0.9°. **c**) Energy versus momentum dispersion collected in the same experimental conditions of (Extended Data Fig. 1a) showing a very symmetric character. To better appreciate this, we extracted MDCs and plotted them in panel **d** along with their extracted *k* values.

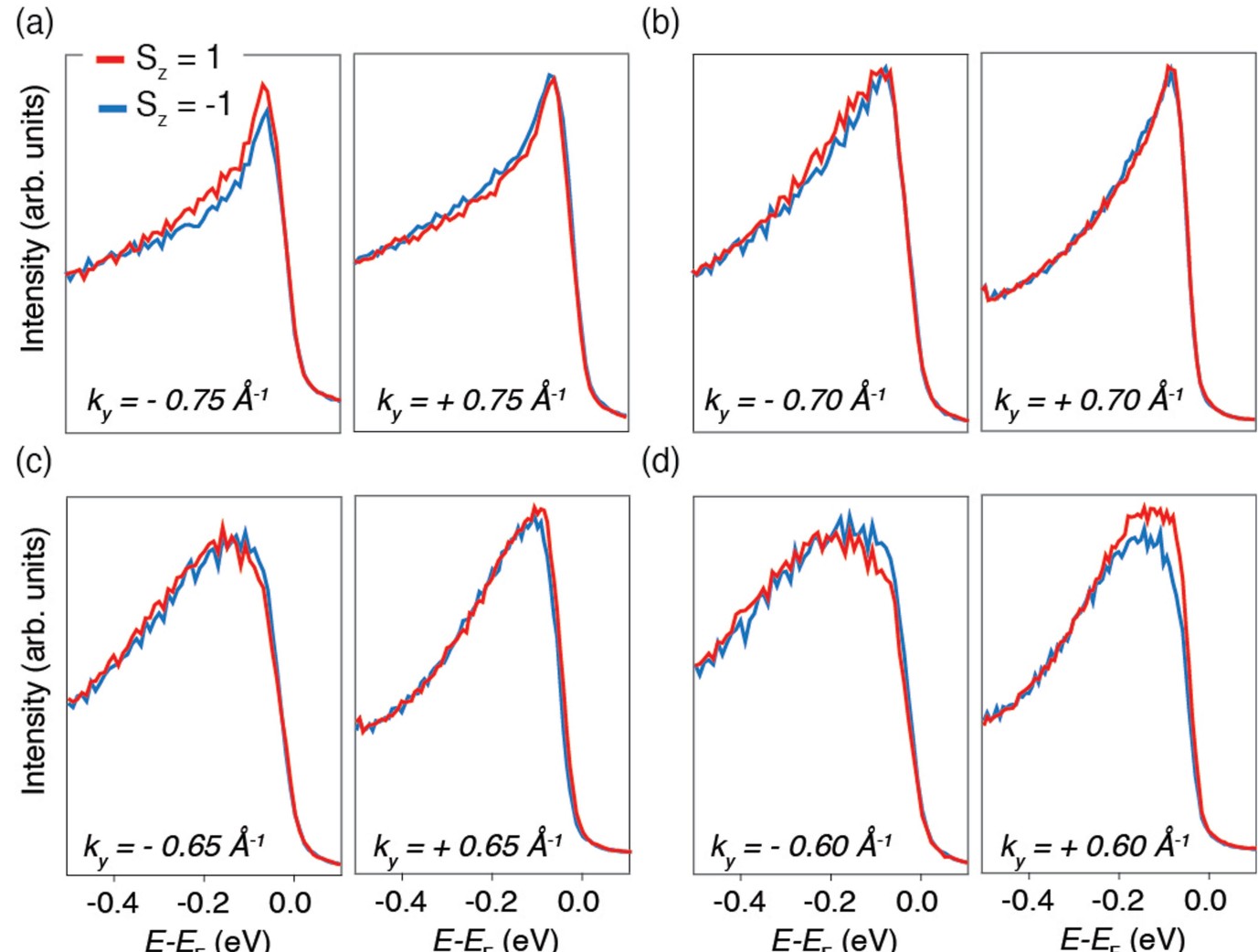

**Extended Data Fig. 3 | Spin-resolved data with unpolarised light.** Energy distribution curves collected at momenta **a-d** $k_y = 0.75\,Å^{-1}$, $k_y = 0.70\,Å^{-1}$, $k_y = 0.65\,Å^{-1}$, $k_y = 0.60\,Å^{-1}$. The data are with sum of circular right and left light and spin-up and spin-down channels have been shown in red and blue, respectively.

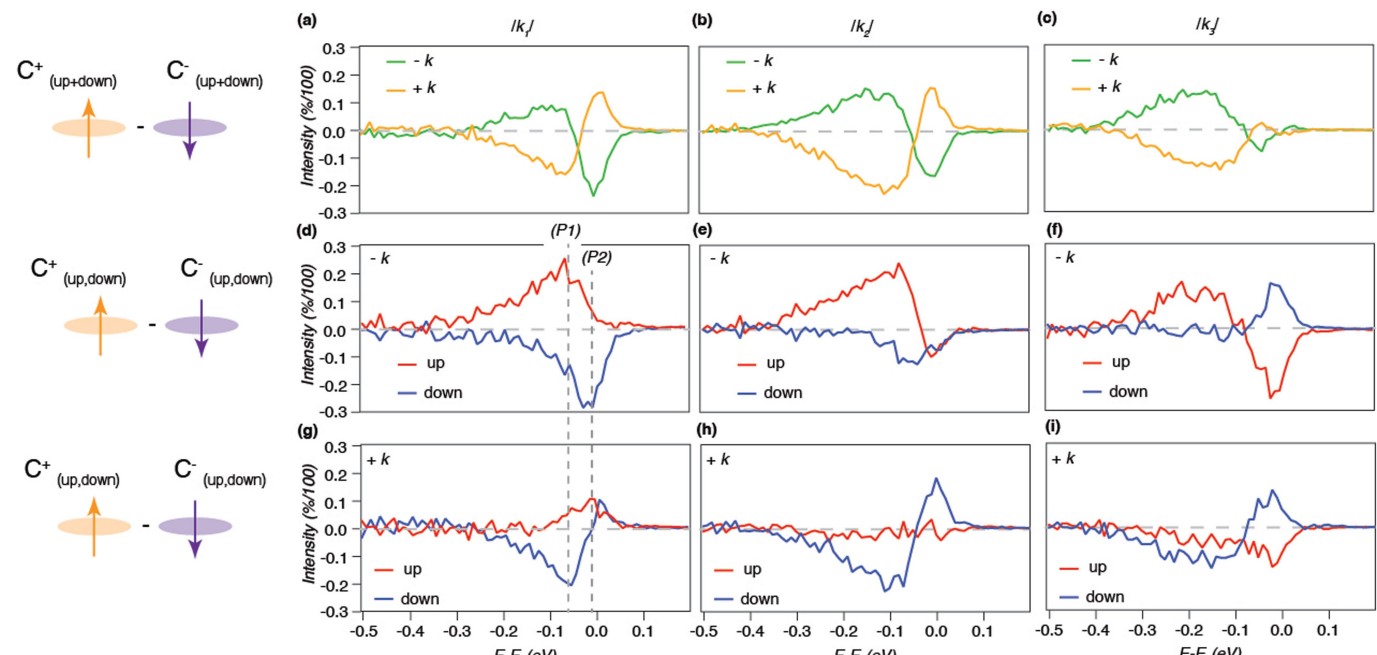

**Extended Data Fig. 4 | Spin-integrated and spin-resolved dichroism.** Spin integrated circular dichroism collected at **a** $k_y = \pm 0.73 \text{Å}^{-1}$ ($k_1$), **b** $k_y = \pm 0.68 \text{Å}^{-1}$ ($k_2$), and **c** $k_y = \pm 0.72 \text{Å}^{-1}$ ($k_3$), as indicated in the main text Fig. 3c. Green curves indicate negative $k$, and orange curve positive $k$. **d-e-f** Spin-resolved circular dichroism collected at negative $k$ for the three momenta indicated. **g-i** Same but collected at positive momenta.

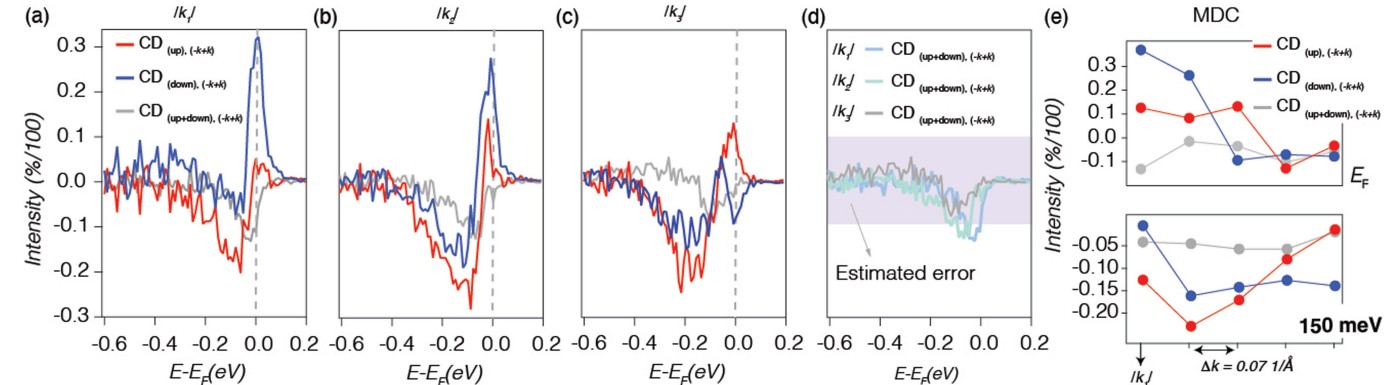

**Extended Data Fig. 5 | Amplitude of the dichroism, EDC and MDC.**
**a-b-c** The amplitudes of the dichroism (at $k$-summed up to see the actual residual) are reported for $k_{1,2,3}$. These show that, while **d** the spin-integrated signal (grey curve) shows a finite value, as large as 10% (which is also very similar to the experimental uncertainty as reported in[39] - purple stripe), the spin-resolved channels indicate a significantly larger amplitude, of a factor larger than × 2 and × 3 for up and down channels, respectively. **e** The amplitude of the dichroism have been also collected by using MDC at two binding energies, i.e., at the Fermi level and at 150 meV below it. As one can see, the grey line, which is the spin-integrated dichroism is nearly flat (in average is 7% - obtained by summing up all the points), while the spin-up and spin-down channels are varying and well-different. The fact that these are also varying is quite remarkable and indicates that our signal is intrinsic in nature.

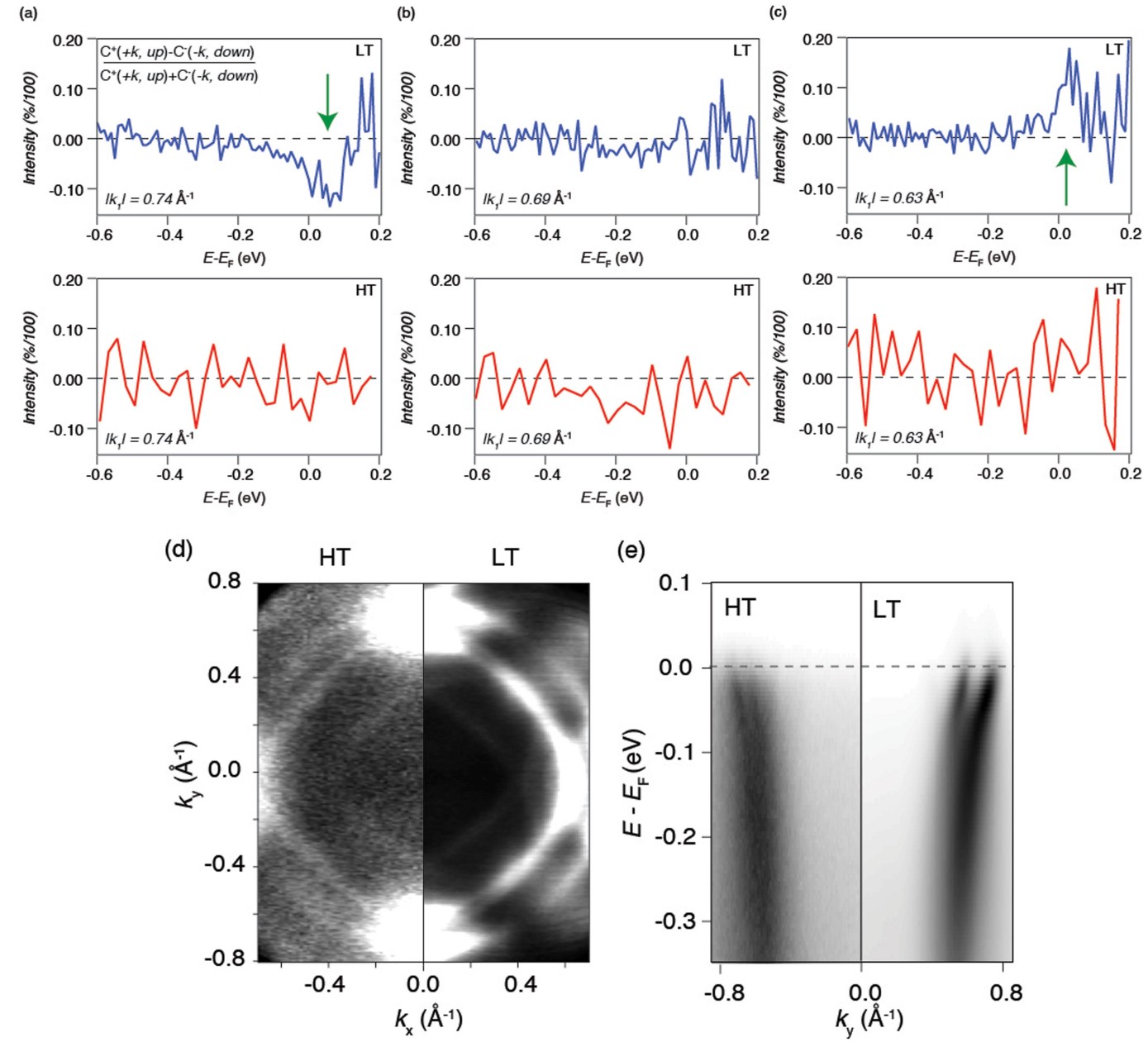

**Extended Data Fig. 6 | Temperature dependence data. a-c** The upper line (blue curves) shows the difference between $C^+(+k, \uparrow)$ and $C^-(k, \downarrow)$ - normalized by their sum - at three values of $k$ and at low temperature (LT), while the lower line is the same for the data collected at 70 K (high temperature, HT), above the magnetic transition. When at LT we do observe a varying finite signal, it starts from negative and it switches sign into positive as a function of $k$, at HT we did not see such a variation. **d** and **e** show the Fermi surface maps and energy versus momentum dispersion for both LT and HT data. The surface states are visible in both cases.

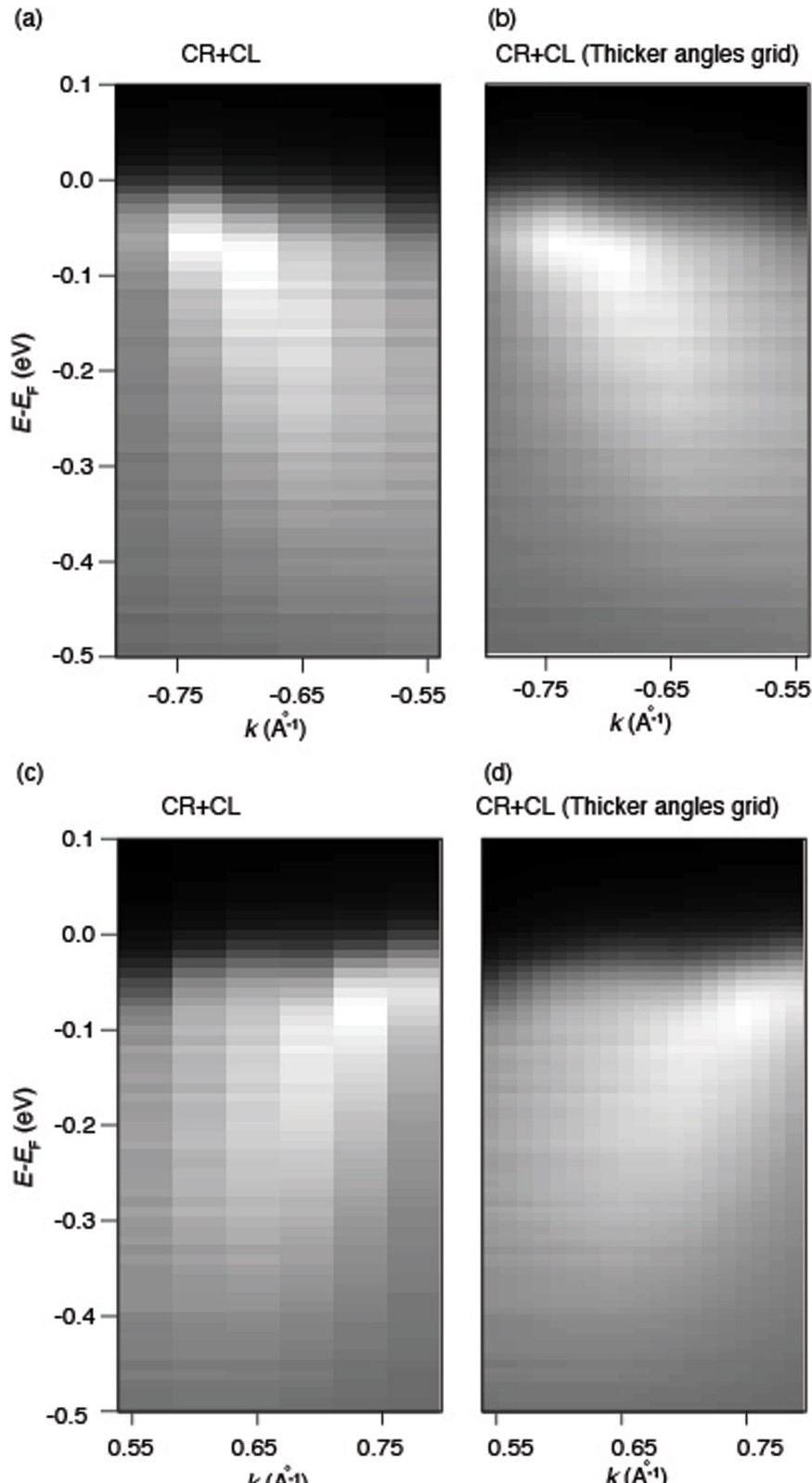

**Extended Data Fig. 7 | Example of some ARPES from the spin-detector.**
**a** Energy dispersions as a function of negative $k$-values obtained by using the spin-detector only in a course alignment scan. **b** Same as **a** but interpolated for a thicker angular grid to see the bands better. **c** Energy dispersions as a function of positive $k$-values obtained by using the spin-detector only in a course alignment scan. **d** Same as **c** but interpolated for a thicker angular grid to see the bands better.