## [Peer Review File · Nature]

Manuscript Title: Signatures of a surface spin-orbital chiral metal

Reviewer Comments & Author Rebuttals

Reviewer Reports on the Initial Version:

Referees' comments:

Referee #1 (Remarks to the Author):

Mazzola et al. present an ARPES study of Sr₂RuO₄ focussing on a combination of circular dichroism and photoelectron spin detection. The authors claim that their data reveals a subtle mirror symmetry breaking in the electronic structure consistent with the existence of spin-orbital currents. In particular, their theory predicts a state with mirror-symmetric orbital moment but mirror-symmetry-broken spin-orbital moments, which, so the authors, is confirmed by their dichroic spin-resolved data. The authors also claim „discovery“ of a new methodology, based on the combination of dichroism and spin detection, which provides „the first experimental avenue to detect chirality-derived symmetry breaking.“

As I elaborate below, the manuscript contains serious and obvious technical errors, which invalidate the conclusions. The manuscript is therefore not suitable for publication.

The key observation, that the authors claim to infer from their data, is that the spin-integrated dichroism does not show mirror-symmetry breaking, i.e. $CD(k) = -CD(-k)$ within experimental uncertainty (Fig.2), while the spin-resolved dichroism does (Fig. 3). However, this claim is invalidated by their own data. A closer look at the CD line cut in Fig. 2b shows that the absolute value of the CD signal at the respective peak extrema is ca. 10 % larger at positive k when compared to negative k. It is also easily seen that the positions of the CD maxima/minima are not symmetric. Along positive k the positions of the maxima/minima appear at smaller absolute k values than along negative k. Similar deviations between $CD(k)$ and $-CD(-k)$ are also seen in Fig. 2c. So, in fact, contrary to the authors' statement, in their data $CD(k)$ is not equal to $-CD(-k)$. Instead there are deviations beyond experimental uncertainty. The deviations are similar or even larger in magnitude compared to the ones observed for the spin-resolved signals in Fig. 3. Irrespective of their origin, the clear experimental deviations between $CD(k)$ and $-CD(-k)$ are incompatible with the claims, the theoretical interpretation and the conclusion of the manuscript. It is confusing to me that the authors claim a „perfectly opposite dichroism at opposite momenta“, while this obviously contradicts their own data.

While the deviations between $CD(k)$ and $-CD(-k)$ could indicate that already the spin-integrated signal reflects an intrinsic mirror-symmetry breaking, the much more likely scenario is that the small asymmetries arise from a not perfectly symmetric experimental setup or sample misalignment. The latter is corroborated by the fact that even the measured band dispersion is not symmetric along +k and -k. This is seen, e.g., in Fig. 1c, where, looking at the highest binding energy, the band along

negative k appears at slightly larger absolute k than along positive k .

As a general remark, I believe that the demonstration of broken mirror symmetry in a spin-resolved/dichroic ARPES experiment requires a considerably more careful experiment than the one presented by the authors in their manuscript. If there is knowledge about the temperature dependence of the magnetic state, it might be possible to demonstrate the absence of mirror-symmetry breaking above the transition temperature, in order to exclude that small misalignments cause the asymmetries observed in the supposed symmetry-broken state at a lower temperature.

Referee #2 (Remarks to the Author):

The purpose of this manuscript is to establish a new way of probing exotic electronic ordering, especially those involving chiral loop currents. The method introduced here uses angle resolved photoemission with circular light polarization and spin detection. Such electronic orders are indeed invoked to explain the behavior of complex materials (from cuprates to the recent kagome metals), but are very difficult to detect. New, well controlled, experimental probes are highly desirable. Therefore, I consider that the subject of this manuscript is relevant for a journal like Nature. Whereas the combination of spin resolution and circular polarization is not new (some references could be added, e.g. PRL 78, 1126, PRL 112, 127002...), it is the first time to my knowledge that it is proposed in this context.

The manuscript is organized in 3 parts : (i) an introduction of the problem and description of the different types of orders and symmetries that will be considered, (ii) an experimental study of a subtle symmetry breaking at the surface of Sr₂RuO₄ (proposed before by μ SR (ref. 7) and STM (ref. 40) experiments) and (iii) a model calculation of the signals expected in case of orbital or spin-orbital quadrupole currents. This study is challenging in many ways. First, the observed effect (essentially the difference between orange and green signals in Fig. 3) is of course very small and great care has to be taken to ensure it is not an artefact. Second, the discussion is relatively complicated to follow, as the states considered here are less intuitive than conventional spin or orbital order.

In a nutshell, the authors demonstrate that :

(1) L_z component alone obeys TRS and mirror symmetry (Fig. 2).

(2) The LS combination does not (Fig. 3).

From their model in part (iii), this is consistent with spin-orbital currents and not with orbital currents, hence adding useful understanding of the possible nature of the chiral currents. I have the following questions :

- How do they rule out from this measurement that this is a pure spin effect ? I am a little bit confused of what they mean by Fig. S3 in supplementary. I suppose they consider that there is no net spin effect at one k value (although in some cases like $k_y=0.6$, the effect is not very different from Fig. S2). If yes, the inequivalence of spin amplitude at k and $-k$ is not really easy to get from this figure. Could they clarify this ?

- At first sight, the signals in Fig. 2c are not really opposite to each other. Could they show this explicitly and explain what they consider as uncertainty of the measurement.

- How would a misalignment show up in each of the case ? What is their quantitative criteria to rule out this plays a role here ?
- Would it be possible to perform the study at higher temperature (above 50K) to check whether the effect disappears/weakens as suggested in the muSR paper (ref. 7) ?

The paper is quite well written, but I think a few points could be improved.

- I find that the « perspective » view of the ARPES images is most of the time not really helpful. I suggest to show them in the regular way. Even in Fig. 1c where it is important to distinguish the mirror plane of light incidence from the axis of measurement, I do not find it very clear.
- In the third paragraph (line 58-75) and Fig. 1b, it is not clear what is just a reminder of general behavior and what will be a finding of the paper (line 72-75). In the bottom part of Fig. 1b, are the sketches of orbital and spin the ones of the spin-orbital current at these 2 opposite k values ? How are they chosen and do they obey TRS ?
- I do not understand what the authors mean line 116 by : «upon saturation ».

Referee #3 (Remarks to the Author):

In the present work, the authors show evidences for the presence of chiral currents on the surface of the enigmatic material Sr₂RuO₄ and connect experimental findings with a theoretical description of the electronic structure in that state. This state is argued to be a novel phase of matter that has connections to other proposals of loop currents in quantum materials. The experimental setup is based on photoelectron spectroscopy where circularly polarized light is used and the photoelectrons are detected with their spin quantum number allowing to reveal the spin-angular momentum as the authors argue. In summary, an intriguing new finding for the quantum material Sr₂RuO₄ that opens the possibility to reconcile a number of other experimental findings in this system.

The referee believes that the present work has the potential to be highly influential and introduces a new concept together with experimental methods and could be suitable for publication in Nature in principle. However, two shortcomings should be explained in detail and incorporated into the manuscript before the referee can make a final evaluation and provide guidance whether the work should be accepted for publication. First, the authors should work out the concept about the chiral metal to a degree that it is understandable to the non expert without need of background information Second, there are some open questions regarding the theoretical description, the significance of the surface reconstruction in this material and the connection between measured quantity in the ARPES experiment and the theoretical modeling and material-specific properties.

Here come some point-by point questions and suggestions to improve the manuscript in the order as they occur in the text:

1. In the abstract, the authors write about “develop a new theory”. Indeed, there is a microscopic Hamiltonian and the calculation of the angular momentum operator presented in the work. However, the direct link between the electronic structure exhibiting loop currents and the experimental measurements, i.e. the coupling of light to the electronic states and the expected measured quantities could be presented more clearly.

2. The authors stress in the abstract that the data is compatible with “quadrupole currents at the surface”. Is there any experimental evidence (from the current study) that the currents only exist on the surface? Or is it merely an indirect conjecture because bulk sensitive measurements have so far not detected any evidence of these currents yet.
3. page 3, lines 50-70: The authors try to describe the physical consequences of spin and orbital textures in the electronic structure. However, there are a number of concepts introduced and used that would make it difficult for a non-expert reader to understand the essence. The wording could be expanded and made more accessible by illustrating with (known) examples and/or a mathematical description of symmetry properties might be needed.
4. page 4, line 85: Similar comment here. What does “profile of LS quadrupole texture” mean here? Indeed, the abbreviation LS is defined, but that quantity should be described better for example by defining it mathematically.
5. page 4, line 91: The authors use the experimental method of CP-Spin ARPES. The referee believes that some additional discussion is needed to make the reader understand how the spin and orbital moment can be detected with this method.
6. Fig. 1: panel (a) Is there a typo for the “Spin” example? Should it read $\langle S_i \rangle \neq 0$ instead of $\langle S_{ij} \rangle$? What about the possibility of “spin orbital”, i.e. $\langle S_i \rangle \neq 0$ and $\langle L_i \rangle \neq 0$ (but not the quadrupole moment? Can such a current be created? Does it have distinct properties from just a superposition of spin and orbital currents?
7. Figure 4 (a): As understood, this is a plot of the spectral function for the surface electronic structure. In detail, this looks quite different from (a) other theoretical models and (b) other ARPES results on SRO for example the work by Tamai et al. (Ref. 38) Fig. 7 (right). Please explain.
8. Figure 4 (b): Can you please explain the significance / meaning of this panel. Is this supposed to be a real space sketch of the currents. If yes, why are the currents running along the (1,1,0) direction and not along (1,0,0) for example as it would be expected for currents carried by the nearest neighbor hopping. How should one understand these currents? Are these supposed to be loop currents that close with a length scale? (What would be this length scale?)
9. page 11, line 180: The authors argue to ignore the surface reconstruction which is supposed to go together with the octahedral rotation. Certainly, the octahedral rotations also break a number of mirror symmetries and introduce handedness i.e. chirality in the (static) electronic structure. Why does this not alter the conclusions / analysis? Could this be the origin of the breaking of the symmetry in the ARPES measurements?
10. page 11, line 187: The authors state that the mentioned expectation values are “related to the dichroic and spin dichroic amplitudes probed by ARPES”. Can this connection be made a little more precise? What does “related” mean here? Can one see this from the coupling to the vector potential? Additional related question: Are also matrix elements of the dipole operator involved which can have nontrivial effects if mirror symmetry of the orbital states (Wannier states for example) is broken as it happens on the surface?
11. references: Minor comment. Note that “Titles of all cited articles are required” by style of the journal. Having the titles already on review stage would make the reading of manuscript and evaluation of literature easier.
12. Fig. 1 (Supplement): As understood, panel (a) should show the spectral function of the surface states. Apologies for the ignorance here, but it is difficult to see similarities of this spectral function with the theoretically calculated one in Fig. 4 (a). Please comment and point out the important features. Also, the data seems to deviate from the measurements in Ref. 38 and for example Phys.

Rev. Lett. 110, 097004 (2013), Chinese Phys. Lett. 29 067401 (2012), Phys. Rev. B 86, 165112 (2012).

13. Supplement, page 3, line 56: The authors write “the bulk of Sr₂RuO₄ are unpolarized in spin”. Is this a conclusion from the current experimental data, or a summary from other measurements in the literature. If the latter is true, please indicate that and underline with literature citations.

14. Supplement, page 6, eq. (3): Is there any particular significance of writing the components of the normal state Hamiltonian by help of the L_i matrices? This seems just a rewriting of the model in Ref. 1.

15. Supplement, page 7, line 89: A small Rashba spin orbit coupling term is introduced. Indeed, such a term is expected by symmetry and as the authors argue, experimentally this term is bounded to be small. Question: Do the conclusions regarding the calculated expectation values of the spin and orbital moments or other quantities change if this term is set to zero?

16. General question about coordinate system: Is there some inconsistency between several parts of the manuscript? The experimental data is analyzed along the momentum path $X \rightarrow \Gamma \rightarrow X'$ (see Fig. 3, c). However, the spin and orbital moments are calculated along $M'' \rightarrow \Gamma \rightarrow M$. Please clarify.

Author Rebuttals to Initial Comments:

Report Referee 1

Mazzola et al. present an ARPES study of Sr_2RuO_4 focussing on a combination of circular dichroism and photoelectron spin detection. The authors claim that their data reveals a subtle mirror symmetry breaking in the electronic structure consistent with the existence of spin-orbital currents. In particular, their theory predicts a state with mirror-symmetric orbital moment but mirror-symmetry-broken spin-orbital moments, which, so the authors, is confirmed by their dichroic spin-resolved data. The authors also claim discovery“ of a new methodology, based on the combination of dichroism and spin detection, which provides the first experimental avenue to detect chirality-derived symmetry breaking.“

As I elaborate below, the manuscript contains serious and obvious technical errors, which invalidate the conclusions. The manuscript is therefore not suitable for publication.

We respectfully disagree with the Referee’s criticisms regarding technical errors in our manuscript. However, we appreciate their thorough review, as it provided us with the chance to enhance the clarity of our argument and improve the overall quality of our work. We took their comments seriously and conducted additional theoretical analysis and measurements, as requested, to support the validity of our study and confirm our conclusions. To address the Referee’s concerns, we will respond to each comment in detail.

The key observation, that the authors claim to infer from their data, is that the spin-integrated dichroism does not show mirror-symmetry breaking, i.e. $\text{CD}(k) = -\text{CD}(-k)$ within experimental uncertainty (Fig. 2), while the spin-resolved dichroism does (Fig. 3). However, this claim is invalidated by their own data. A closer look at the CD line cut in Fig. 2b shows that the absolute value of the CD signal at the respective peak extrema is ca. 10 % larger at positive k when compared to negative k . It is also easily seen that the positions of the CD maxima/minima are not symmetric. Along positive k the positions of the maxima/minima appear at smaller absolute k values than along negative k . Similar deviations between $\text{CD}(k)$ and $-\text{CD}(-k)$ are also seen in Fig. 2c. So, in fact, contrary to the authors’ statement, in their data $\text{CD}(k)$ is not equal to $-\text{CD}(-k)$. Instead there are deviations beyond experimental uncertainty. The deviations are similar or even larger in magnitude compared to the ones observed for the spin-resolved signals in Fig. 3. Irrespective of their origin, the clear experimental deviations between $\text{CD}(k)$ and $-\text{CD}(-k)$ are incompatible with the claims, the theoretical interpretation and the conclusion of the manuscript. It is confusing to me that the authors claim a perfectly opposite dichroism at opposite momenta“, while this obviously contradicts their own data.

The referee’s observation regarding the asymmetry of the dichroic signal in Fig. 2c is correct. We believe that it is important to elaborate on this point and report a real amplitude difference in % for both the spin-integrated and spin-resolved signals. This helps to quantify the symmetry/asymmetry and demonstrates that the spin-resolved dichroic signal is significantly more asymmetric than the spin-integrated signal. By doing so, we can also explain why the peak position seems to be slightly off, which is purely intrinsic to the asymmetry itself.

We can begin by discussing the relative amplitudes for the spin-integrated versus spin-resolved signals, as a 10% uncertainty in the measurements is expected. In another recent paper (Di Sante et al. Nature Physics, 2023 - <https://www.nature.com/articles/s41567-023-02053-z>), we found that the dichroic residual was about 8% when measured at the exact point, where geometrical matrix elements are zero by definition. We attributed the oscillation to experimental errors, but to en-

sure we do not exclude any other sources of asymmetry, we continued from an experimental point of view. We found that although the referee identified an asymmetry in the dichroic signal, the spin-resolved data showed a much larger one. We plotted the relative amplitudes of $+k$ and $-k$ of the circular dichroism resolved in the spin channels and also integrated. It is important to note that the integrated signal here was obtained for the spin-resolved data (we summed up spin-up and spin-down channels) to ensure consistency in terms of normalization, analysis, and data treatment.

The discussion has been covered in great detail in both the main text and the supplementary information. Figure 1 displays the results of the amplitude analysis that was described earlier. Panel "a" shows a small asymmetry in the circular dichroism, which is spin-integrated as depicted in the cartoon image. This asymmetry is estimated to be around 10%, and it can be better seen by examining the gray curve in panel d, which is obtained by adding up the $+k$ and $-k$ contributions. This value is slightly larger than the estimated experimental error and calibration of the dichroism, and it was observed in this work as the range of uncertainty. The variation observed is about the same size as the pink error stripe in Fig. 1, as we will also show theoretically. We performed additional and more thorough calculations that indicate there may be a small anisotropy in the spin-integrated dichroic signal, but it is still much smaller than the spin-resolved ones. The experiment also confirms this, as the spin-resolved amplitude is significantly larger than the spin-integrated one. The time-reversal symmetry breaking and the inequivalence of the spin-resolved amplitude are reflected in the deviations observed. The small energy shift is intrinsic and occurs at the same k , and it is attributable to a non-equivalence of the spin-scattering. Such a spectral weight anisotropy can still be evident in Fig. 1 a, even though time-reversal symmetry is broken.

In addition, even if the spin-integrated dichroism is small, there might be still some asymmetry in it due to intrinsic reasons related to the character of the chiral electronic ordering. We thank the referee for asking to deepen this issue that we have thoroughly addressed in the revised version of the manuscript. Indeed, we have considered the possibility of having an amplitude modulation of the spin-orbital quadrupolar currents that follows the surface reconstruction due to the rotation of the RuO_6 octahedra. This is relevant for the examined physical case as it is known that the Sr_2RuO_4 has a reconstruction at the surface due to the alternate rotation of the octahedra around the c -axis. Then, it is plausible to expect that ground state hosts chiral spin-orbital quadrupole currents which are not spatially homogeneous, being affected by the structural change of the octahedra, that result into different current amplitudes when linking ruthenium sites with inequivalent octahedral rotation (see Fig. 2). We have analyzed this configuration schematically shown in Fig. 2 as a function of the current amplitude unbalance associated with the two sublattices. For such current pattern we find that the band resolved dichroic asymmetry in the reconstructed electronic states (Fig. 3 a) is not exactly zero (Fig. 3 b) as found for the uniform current configuration. In order to get an estimate of the difference between the orbital and spin-projected orbital moments as a function of the current sublattice unbalance, we have followed their integrated values (i.e. summing up all the band contributions and integrating along the $\Gamma - X$ line). The outcome is reported in Fig. 3 b where one can clearly see that a nonvanishing orbital moment is induced by the spatially modulated chiral currents. Nevertheless, the asymmetry for the orbital moment turns out to be much smaller than that related to the spin projected one. For instance, for a spatial modulated of the spin-orbital current with a sublattice unbalance of around 3% we find that the integrated value of the spin-projected orbital moment is about five times larger than that of the orbital moment.

Figure 1: **Amplitude of dichroism.** **a** Spin-integrated signal of the dichroism plotted for $+k$ (green curve) and $-k$ (orange curve). **b** Circular dichroism collected at $-k$ and separated in channels up (red curve) and down (blue curve). **c** Circular dichroism collected at $+k$ and separated in channels up (red curve) and down (blue curve). Importantly, one can see both in **b** and **c** that the two spin-channels, even if collected at the same k value occur with a shift of the spectral weight, due to the asymmetry of the spin-value itself. As we discussed in the main text, this is also an indication of the magnetism of the system, consistent with a time-reversal symmetry breaking. **d** The amplitude of the circular dichroism have been plotted and compared: the gray curve is the spin-integrated circular dichroism amplitude obtained by summing the curves in **a**. One can see that this signal carries an intrinsic asymmetry of about 10%. This value is just a bit larger than what we estimated to be the experimental dichroic asymmetry of the apparatus, described by Di Sante et al. in a previous work and indicated by the pink stripe (Nature Physics 2023, <https://www.nature.com/articles/s41567-023-02053-z>). The blue (red) curve is the amplitude obtained by subtracting the circular dichroism with spin-down (up) channel at $+k$ and $-k$. One can see that both spin-resolved channels are significantly larger than the spin-integrated amplitude. For the spin-up, the amplitude is a factor $\times 2$ larger, for the spin down more than a factor $\times 3$. The inequivalence of the spin-resolved amplitude reflects also the time-reversal symmetry breaking.

Figure 2: **Schematic of the current pattern for uniform and staggered octahedral configurations and intra-octahedra loop currents.** (a) Sketch of the considered broken symmetry state with the arrows indicating a uniform current flowing along the Ru-Ru [110] direction. (b) Sketch of the currents in the presence of A and B sublattices associated with the surface reconstructed octahedral rotations. Orange(blue) arrows indicate currents with different amplitudes connecting ruthenium sites belonging to the $A(B)$ sublattice, respectively. (c) A representative pattern of intra-octahedra loop-current that can yield an effective dominant current flow along the Ru-Ru [110] direction. As one can see, the effective Ru-O-Ru current amplitude tends to vanish because there are two direct opposite currents on the Ru-O and O-Ru adjacent bonds. Instead, when considering the [110] Ru-Ru connection there are currents flowing on the Ru-O_x, O_x-O_y and O_y-Ru bonds that can contribute to make an effective non-vanishing bond current when considering the spin and orbital degrees of freedom in the $d-p$ hybridization and the spin-orbit coupling. Here, O_x and O_y are the planar oxygens along the [100] and [010] directions, respectively.

The point here is that, as discussed in the paper, the spatially homogeneous chiral spin-orbital current would lead to exact zero asymmetry in the dichroic amplitude due to the balance among the torques arising from the chiral current and that one arising from the spin-orbit coupling. Now, since the spin-orbit coupling at the Ru site is homogeneous, a spatial dependent amplitude of the chiral spin-orbital current cannot be compensated and a non-vanishing amplitude of the dichroic signal can occur. Due to the surface reconstruction of the electronic states in the Sr₂RuO₄ one might argue that small deviations of the dichroic signal from zero can be consistent with the theoretical prediction when considering the orthorhombic configuration due to the staggered octahedral rotations around the c -axis. The main observation is that if the chiral spin-orbital current follows the structural reconstruction at the Sr₂RuO₄ surface one can also account for an amplitude asymmetry of the dichroic signal keeping however a significantly larger spin-dichroic asymmetry as compared to the dichroic one. We have now discussed this issue in the main text. We have also added a full dedicated section in the supplementary information with the new outcomes of the theoretical analysis as related to the behavior of the spin-orbital chiral current phase in the presence of surface reconstruction.

While the deviations between $CD(k)$ and $-CD(-k)$ could indicate that already the spin-integrated signal reflects an intrinsic mirror-symmetry breaking, the much more likely scenario is that the small asymmetries arise from a not perfectly symmetric experimental setup or sample misalignment. The latter is corroborated by the fact that even the measured band dispersion is not symmetric along

Figure 3: **Fermi contour map in the presence of surface reconstruction due to the staggered octahedral rotation. Momentum and band averaged asymmetry of the orbital and spin-projected orbital moment, evaluated along the $\Gamma - Y$ direction.** (a) Reconstructed Fermi surface. (b) k -integrated and band-averaged orbital moment $\langle \delta L_z \rangle = \frac{1}{12} \sum_n \int |L_z(n, k) + L_z(n, -k)| dk$ and $\langle \delta L_z^\sigma \rangle = \frac{1}{12} \sum_n \int |L_z^+(n, k) + L_z^-(n, -k)| dk$ assuming an amplitude modulation of the spin-orbital current so that the spin-orbital quadrupole current is given by $J_+^{so} = J_{so} + J_{so}^{stg}$ when connecting Ru sites in octahedra that are clockwise rotated and $J_-^{so} = J_{so} - J_{so}^{stg}$ for anticlockwise rotated octahedra.

$+k$ and $-k$. This is seen, e.g., in Fig. 1c, where, looking at the highest binding energy, the band along negative k appears at slightly larger absolute k than along positive k .

We have checked this and we can safely exclude a significant misalignment in the experiment. The asymmetry in Fig. 1c is probably a problem of the 3D rendering of the graphics, in fact, by extracting the momentum distribution curves (MDC) we cannot see any asymmetry at all, as well as when the data are plotted frontally. To prove this, we have both plotted now the momentum distribution curves and extracted (from fits) the corresponding k values and also fitted the Fermi surface map without adjusting for any azimuthal angle (Fig. 4 a). As the referee can notice, also in the data taken as they are and k -warped, there is no a considerable misalignment. Actually, this is very small and estimated to be maximum 0.9 degrees (Fig. 4 to see the rotated Fermi surface map), thus this is far too small to be causing the anisotropy observed and within the motors angular resolution (and spin-detector resolutions). We report in Fig. 4 the results of the analysis. This has been also presented in a dedicated section of the Supplementary information. After carefully analyzing the standard energy versus momentum maps, we plotted them without any shearing to ensure the absence of significant misalignment. There is no difference between $+k$ and k , and the values extracted by fitting the MDCs in Fig. 4 c indicate that the difference is smaller than the resolutions. We believe that the plot with 3D rendering and shearing did not accurately represent this, but it is clear that there is no k misalignment within the uncertainty of the measurement. We have added this discussion about the rotation and misalignment in the supplementary information, as we believe it is a crucial point. Additionally, we have included a discussion about the fits, which were

Figure 4: **Sample alignment.** In order to carefully perform the measurements, the sample has been aligned along the mirror plane. **a** k -warped Fermi surface map without any azimuthal angle adjustment and fit to the bands (red 'plus'). As one can see the samples is already aligned very well with the interested direction along the analyser slit. Also the edge of the detector is shown to better see that the data is taken in this configuration. In order to make the azimuth the best we could achieve, we would need to **b** rotate the samples not more than 0.9 degrees, which is within the human error and motors resolution in the laboratory. **c** Standard energy versus momentum plot shown frontally to better appreciate the alignment. One can see that by extracting the MDCs (we show this for a range of 200 meV) no difference in k positions for the peaks is observed.

performed using Lorentzian curves. Finally, in the supplementary information, we added a section which describes the V-LEED angular calibration to show how $\pm k$ values were checked, chosen, and monitored for the spin-measurements, without the risk of any angular offset and/or misalignment. We appreciate the Referee's input, as this analysis can provide additional trust to our results.

We have now plotted the spectra also in this fashion and added this discussion about the rotation and about the misalignment in the supplementary information, because we thought that this constitute a very important point. Plus we added the discussion about the fits, which have been performed by using Lorentzian curves. We thank again the Referee as we think that this analysis can truly give additional trust to our results.

As a general remark, I believe that the demonstration of broken mirror symmetry in a spin-resolved/dichroic ARPES experiment requires a considerably more careful experiment than the one presented by the authors in their manuscript. If there is knowledge about the temperature dependence of the magnetic state, it might be possible to demonstrate the absence of mirror-symmetry breaking above the transition temperature, in order to exclude that small misalignment cause the asymmetries observed in the supposed symmetry-broken state a lower temperature.

We thank the referee for this remark. Together with the geometrical configuration and raw alignment shown, we have also performed the measurements above the transition temperature (about 50 K according to the muons spectroscopy) of the surface electronic ordering. The sample was cleaved at 70 K (same sample and orientation) and the surface states, despite much weaker in intensity as compared to the low temperature data were still observed. One can directly see this in Fig. 5 d-e, where the surface states are still visible but much less intense and the bands in the high temperature data appear significantly broader, due to both thermal broadening and also temperature effects on the correlations. The cleave, however, was successful and the samples looked perfectly shiny and mirror-like. By performing the circular-dichroism measurements as in Fig. 3 of the main text, we did observe a significant suppression of the signal which reduces to noise oscillations around zero. Thus, as the referee suggested, the effect visible at low temperature is not observed at 70 K. It is also important to stress that there might be effects due to short-range magnetic fluctuation, however, these are not visible within the resolution of our data and the signal is certainly reduced significantly compared to the data collected at low temperature. To make the analysis more comparative, also here, we have plotted the amplitude of the signals: We compared the results shown in Fig. 3 of the main text ($C^+(+k, \uparrow)$ vs $C^-(-k, \downarrow)$) which were obtained for 16 K with those obtained at 70 K, well above the magnetic transition. As one can see, going from left to right of Fig. 5 (from larger k to smaller k), for temperatures below the transition one, the signal goes from negative (top panel a), it reduces to zero (top panel b), and flip sign to positive (top panel c), going from -15% to 15%. Instead, for the measurements collected above the transition, the signal is reduced to noise around the zero value for all momenta. Note that the large value of the oscillations is also due to the fact that we are normalizing by the sum of the signals. Thus, close to the zero value we are dividing by a very small value. We reported this analysis in the supplementary information and discussed the results in a dedicated section. We have also referred to this effect in the main text.

We thank again the referee for the comments that helped to improve and clarify several points

Figure 5: **Low and High temperature amplitudes of the dichroism.** Here, we compared the amplitudes of the graph shown in Fig. 3 of the main text for low temperature and high temperature data. The top row (blue curves) are the amplitudes for the low temperature signal, whilst the bottom row is for the high temperature one. **a**, **b**, and **c** are the different k values as shown in the main text, namely, 0.74 \AA^{-1} , 0.69 \AA^{-1} , and 0.63 \AA^{-1} . As one can see, going from left to right (from larger k to smaller k), for temperatures below the transition one, the signal goes from negative (top panel a), it reduces to zero (top panel b), and flip sign to positive (top panel c), going from -15% to 15%. Instead, for the measurements collected above the transition, the signal is reduced to noise around the zero value for all momenta. **d** Fermi surface map comparing the high temperature and the low temperature data and **e** corresponding energy versus momentum maps. The surface states are still visible but much less intense and the bands in the high temperature data appear significantly broader, due to both thermal broadening and also temperature effects on the correlations. The cleave however, was successful and the samples looked perfectly shiny.

of the presentation and we believe that we have now addressed satisfactorily all the concerns.

Report Referee 2

The purpose of this manuscript is to establish a new way of probing exotic electronic ordering, especially those involving chiral loop currents. The method introduced here uses angle resolved photoemission with circular light polarization and spin detection. Such electronic orders are indeed invoked to explain the behavior of complex materials (from cuprates to the recent kagome metals), but are very difficult to detect. New, well controlled, experimental probes are highly desirable. Therefore, I consider that the subject of this manuscript is relevant for a journal like Nature. Whereas the combination of spin resolution and circular polarization is not new (some references could be added, e.g. PRL 78, 1126, PRL 112, 127002. . .), it is the first time to my knowledge that it is proposed in this context.

We appreciate the positive remarks of the referee in pointing out that our study is relevant for a journal like Nature and in underlying the originality about the way we are exploiting the combination of spin resolution and circular polarization to detect exotic electronic orderings that are marked by chiral currents phases.

The manuscript is organized in 3 parts : (i) an introduction of the problem and description of the different types of orders and symmetries that will be considered, (ii) an experimental study of a subtle symmetry breaking at the surface of Sr_2RuO_4 (proposed before by muSR (ref. 7) and STM (ref. 40) experiments) and (iii) a model calculation of the signals expected in case of orbital or spin-orbital quadrupole currents. This study is challenging in many ways. First, the observed effect (essentially the difference between orange and green signals in Fig. 3) is of course very small and great care has to be taken to ensure it is not an artefact. Second, the discussion is relatively complicated to follow, as the states considered here are less intuitive than conventional spin or orbital order.

We thank the referee for the valuable observations and for pointing out the challenging problem we are facing. We agree with the referee that the effect has to be analyzed with care by checking that there are no artifacts. For this purpose we have performed various checks.

First, for sake of clarity as well as to explicitly and quantitatively demonstrate the difference in the asymmetry between the dichroic and spin-dichroic signals, we have replotted the data related to Fig. 3 in a way that a direct comparison can be extracted as shown in Figure R1 (d).

Second, we have checked that there are no misalignment issues in the performed experiments.

Finally, as also suggested by the referee, we have analyzed the dichroic and spin-dichroic amplitudes above the transition temperature of the electronic ordering as indicated by muons spectroscopy.

Below, we will elaborate further on these aspects in the point-by-point reply.

In a nutshell, the authors demonstrate that : (1) L_z component alone obeys TRS and mirror symmetry (Fig. 2). (2) The LS combination does not (Fig. 3). From their model in part (iii), this is consistent with spin-orbital currents and not with orbital currents, hence adding useful understanding of the possible nature of the chiral currents. I have the following questions : - How do they rule out from this measurement that this is a pure spin effect ? I am a little bit confused of what they mean by Fig. S3 in supplementary. I suppose they consider that there is no net spin effect at one k value (although in some cases like $k_y=0.6$, the effect is not very different from Fig. S2). If yes, the inequivalence of spin amplitude at k and $-k$ is not really easy to get from this figure. Could they clarify this ?

We are grateful to the referee for the valuable observations and remarks. Concerning the question on the reason why we are ruling out that the measurement is a pure spin effect, we would like to point out that the presence of a magnetic order set out by a periodic arrangement of the local spin moment at the Ru site would also imply a non-vanishing orbital component at each k point in the Brillouin zone and an asymmetry in the amplitudes at k and $-k$ in case of mirror broken configurations. Then, for a nontrivial magnetic ground state that breaks both time-reversal and mirror, the measurement should provide a nonvanishing and comparable in amplitude asymmetry of the signal for both the dichroic and the spin-dichroic channels. Instead, for a mirror symmetric magnetic pattern one would get a vanishing asymmetry for both dichroic and the spin-dichroic signals. For our purposes, in order to quantitatively demonstrate the consequences of a magnetic ground state we have computed, for a representative antiferromagnetic configuration with canted moments, the orbital and spin-resolved orbital moment for all the bands crossing the Fermi level (Fig. 6). As one can see from the inspection of the results in Fig. 6 the orbital and spin-projected orbital moments exhibit a sizable asymmetry when comparing the amplitudes at k and $-k$. This is a general feature of symmetry analogue magnetic phases with spatially ordered Ru spin moments. Instead, the experimental observation indicates that the dichroic asymmetry is substantially smaller than the spin-dichroic amplitude. Thus, we conclude that a magnetic ordering can be ruled out to interpret our findings.

- At first sight, the signals in Fig. 2c are not really opposite to each other. Could they show this explicitly and explain what they consider as uncertainty of the measurement.

We thank the referee for this remark that we have taken carefully into account. The referee is correct in saying that there is an asymmetry in the dichroic signal in Fig. 2c. This is indeed an important point to be further elaborated. To this aim, it is appropriate to report a direct amplitude difference in percentage for both the spin-integrated and the spin-resolved signals. This comparison helps to quantify the symmetry/asymmetry and at the same time, allows to show in a clear way that the spin-resolved dichroic signal is substantially more asymmetric than the spin-integrated one. Indeed, as one can see in panel a of Fig. 1, the circular dichroism (which is integrated in the spin channels as described in the cartoon picture) presents a small asymmetry in a small energy window, which is estimated to be 10%. The asymmetry can be better visualised by considering the gray curve of panel d, which is obtained by summing up the contributions at $+k$ and $-k$. The resulting value is just a little bit larger than the estimated experimental error and calibration of the dichroism. Indeed, as also demonstrated in the paper by Di Sante et al. (Nature Physics, 2023 - <https://www.nature.com/articles/s41567-023-02053-z>), the dichroic signal can be of the same order of magnitude ($\sim 8\%$) even at the Γ point of the Brillouin zone, where, for nonmagnetic materials, it is expected to be identically zero.

Concerning the intensity of the observed asymmetry in the dichroic signal, we have also considered the possibility of having some asymmetry in it due to intrinsic reasons arising from the character of the chiral electronic ordering. In this context, as also suggested by the referees, we have analyzed an electronic ordering that is marked by an amplitude modulation of the spin-orbital quadrupolar currents that follows the surface reconstruction due to the rotation of the RuO_6 octahedra.

To this aim, we have analyzed a chiral electronic state marked by an amplitude modulation of the spin-orbital quadrupolar currents that results from the surface reconstruction due to the rotation of the RuO_6 octahedra (Fig. 2). This is relevant for the examined physical case as Sr_2RuO_4 is known to have a structural reconstruction at the surface due to the alternate rotation of the octahedra

Figure 6: **Orbital and spin-projected orbital moment for a canted antiferromagnetic ground state in the presence of staggered octahedral rotations.** **a** amplitude of the orbital angular momentum $L_z(n, k) = \langle \psi_{n,k} | \hat{L}_z | \psi_{n,k} \rangle$ for each band $|\psi_{n,k}\rangle$ with $n = 1, \dots, 12$ evaluated along the $\Gamma - X$ direction. For clarity we plot both $L_z(n, k)$ and $-L_z(n, -k)$ for any given momentum k to directly compare the amplitudes at opposite momenta for all bands $|\psi_{n,k}\rangle$ evaluated along the $k_y = 0$ direction. **b** amplitude of the spin projected orbital angular momentum $L_z^\pm(n, k) = \langle \psi_{n,k} | (1 \pm \hat{s}_z) \hat{L}_z | \psi_{n,k} \rangle$ evaluated along the $k_y = 0$ direction. For all the bands, both $L_z(n, k)$ and $L_z^\pm(n, k)$ do not show any symmetry and do not match at k and $-k$. The amplitudes of the sublattice magnetizations are $\mathbf{M}_A = \mathbf{M}_{\text{uni}} + \mathbf{M}_{\text{stg}}$ and $\mathbf{M}_B = \mathbf{M}_{\text{uni}} - \mathbf{M}_{\text{stg}}$ with $\mathbf{M}_{\text{stg}} = (0, 0, 0.1)$ and $\mathbf{M}_{\text{uni}} = (0.02, 0, 0)$ in units of $\text{eV}(\mu_B)^{-1}$.

around the c -axis. Then, it is plausible to expect that the chiral spin-orbital quadrupole currents are not spatially homogeneous, being affected by the structural change of the octahedra, that result into different current amplitudes when linking ruthenium sites with inequivalent octahedral rotation (see Fig. 2 (b)). We have analyzed this configuration as a function of the current amplitude unbalance associated with the two sublattices. For such current pattern we find that the band resolved dichroic asymmetry in the reconstructed electronic states (Fig. 3a) is not exactly zero as found for the uniform current configuration. In order to get an overall estimate of the difference between the orbital and spin-projected orbital moments as a function of the current sublattice unbalance, we have determined the corresponding integrated values (i.e. summing up all the band contributions and integrating along the $\Gamma - X$ line). The outcome is reported in Fig. 3b where one can clearly see that a nonvanishing orbital moment is induced by the spatially modulated chiral currents. Nevertheless, the asymmetry for the orbital moment turns out to be much smaller than that related to the spin projected one. For instance, for a spatial modulation of the spin-orbital current with a sublattice unbalance of around 3% we find that the integrated value of the spin-projected orbital moment is about five times larger than that of the orbital moment. The point here is that, as discussed in the paper, the spatially homogeneous chiral spin-orbital current would lead to exact zero asymmetry in the dichroic amplitude due to the balance among the torques arising from the chiral current and that one arising from the spin-orbit coupling. Now, since the spin-orbit coupling at the Ru site is homogeneous, a spatial dependent amplitude of the chiral spin-orbital current cannot be compensated and a non-vanishing asymmetry of the orbital moment can occur. Due to the surface reconstruction of the electronic states in the Sr_2RuO_4 one might argue that small deviations of the dichroic signal from zero can be consistent with the theoretical prediction when considering the orthorhombic configuration due to the staggered octahedral rotations around the c -axis. The main observation is that if the amplitude of the chiral spin-orbital current follows the structural reconstruction at the Sr_2RuO_4 surface one can also account for a small asymmetry of the dichroic signal yielding however a significantly larger spin-dichroic asymmetry as compared to the dichroic one. We have now introduced this analysis in the main text and also discussed this issue in details. We have also added a full dedicated section in the supplementary information with the new outcomes of the theoretical analysis as related to the behavior of the spin-orbital chiral current phase in the presence of surface reconstruction.

We have added a thorough discussion in both the main text and in the supplementary information to discuss in details about the issue of the dichroic asymmetry.

- How would a misalignment show up in each of the case ? What is their quantitative criteria to rule out this plays a role here ?

- Would it be possible to perform the study at higher temperature (above 50K) to check whether the effect disappears/weakens as suggested in the muSR paper (ref. 7) ?

We thank the referee for this suggestion. We have analyzed the dichroic and spin-dichroic amplitudes at a temperature (i.e. 70 K) that is above the transition one for the electronic ordering as indicated by the muons spectroscopy experiment. The sample was cleaved at 70 K (same sample and sample orientation) and the surface states, despite much weaker in intensity as compared to the low temperature data (this behavior was always found for this temperature and thus it is not surprising given the surface nature of the states) were still observed. This is confirmed by the

outcomes of Fig. 5 d-e, where the surface states are still visible though much less intense and the bands in the high temperature data appear significantly broader, due to both thermal broadening and also temperature effects on the correlations. The cleave however, by eye, was successful and the samples looked perfectly shiny and mirror-like. By performing the circular-dichroism measurements as in Fig. 3 of the main text, we did observe a sizable suppression of the signal which reduces to noise oscillations around zero.

We would like to point out that while short-range magnetic fluctuations can contribute, these are not visible within the resolution of our data and the signal is certainly reduced significantly compared to the data collected at low temperature. In order to make the analysis quantitatively comparative, we have now compared the results shown in Fig. 3 of the main text ($C^+(+k, \uparrow)$ vs $C^-(-k, \downarrow)$) which were obtained for 16 K with those obtained at 70 K, above the magnetic transition. As one can see, going from left to right of Fig. 5 (from larger k to smaller k), for temperatures below the transition one, the signal goes from negative, it reduces to zero, and flip sign to positive, going from -15% to 15%. Instead, for the measurements collected above the transition, the signal is reduced to noise around the zero value. It is worth pointing out that the large value of the oscillations is also due to the fact that we are normalizing by the sum of the signals. Thus, close to the zero value we are dividing by a very small value. These results and the corresponding analysis are included in the supplementary information in a dedicated session. We have also commented about the role of temperature in the main text of the manuscript.

The paper is quite well written, but I think a few points could be improved. - I find that the perspective view of the ARPES images is most of the time not really helpful. I suggest to show them in the regular way. Even in Fig. 1c where it is important to distinguish the mirror plane of light incidence from the axis of measurement, I do not find it very clear.

We thank the referee for this suggestion. We decided to follow it and we have now shown the data in a 'standard' way.

- In the third paragraph (line 58-75) and Fig. 1b, it is not clear what is just a reminder of general behavior and what will be a finding of the paper (line 72-75). In the bottom part of Fig. 1b, are the sketches of orbital and spin the ones of the spin-orbital current at these 2 opposite k values ? How are they chosen and do they obey TRS ?

We thank the Referee for this request of clarification. From line 58 to 70 we provide a general discussion of the behavior of dipolar and quadrupolar observables. While from line 70 to 75 we state the main finding of the paper. We have slightly rephrased the paragraph in order to make more evident the difference between the general introduction and the statement of our findings.

In the bottom part of the sketch of Fig. 1b, the scheme indicates a configuration with a spin-orbital quadrupole current flowing along the indicated k -direction. The spin-orbital quadrupole current that is depicted is a mark of the ground state and it is mirror and time-reversal symmetry broken. On the other hand, the orbital configuration that is displayed corresponds to the amplitudes at symmetry related k points. We have further clarified the outcome of the sketch by specifying in the caption that the spin-orbital current refers to the ground state and the orbital moment is the observable at k and $-k$ momenta.

- I do not understand what the authors mean line 116 by : upon saturation.

We apologize this was not clear. We meant that by changing the contrast the states are better

visible. We changed this now and rephrased it to make it clearer.

Report Referee 3

In the present work, the authors show evidences for the presence of chiral currents on the surface of the enigmatic material Sr_2RuO_4 and connect experimental findings with a theoretical description of the electronic structure in that state. This state is argued to be a novel phase of matter that has connections to other proposals of loop currents in quantum materials. The experimental setup is based on photoelectron spectroscopy where circularly polarized light is used and the photoelectrons are detected with their spin quantum number allowing to reveal the spin-angular momentum as the authors argue. In summary, an intriguing new finding for the quantum material Sr_2RuO_4 that opens the possibility to reconcile a number of other experimental findings in this system. The referee believes that the present work has the potential to be highly influential and introduces a new concept together with experimental methods and could be suitable for publication in Nature in principle.

We sincerely thank the Referee for praising our work as highly influential and able to introduce new concepts and experimental methods in such a way to make it suitable for Nature.

However, two shortcomings should be explained in detail and incorporated into the manuscript before the referee can make a final evaluation and provide guidance whether the work should be accepted for publication. First, the authors should work out the concept about the chiral metal to a degree that it is understandable to the non expert without need of background information. Second, there are some open questions regarding the theoretical description, the significance of the surface reconstruction in this material and the connection between measured quantity in the ARPES experiment and the theoretical modeling and material-specific properties.

We thank the referee for asking to clarify these issues that will help to substantially improve the overall general readability and impact of the manuscript. We have thoroughly taken into account these criticisms and accordingly revised the manuscript as also reported in the point-by-point reply below.

Concerning the concept of chiral metal, one can employ the analogy with chiral crystals and their symmetry properties (see, e.g., G. Chang et al., *Nature Mater.* 17, 978 (2018), and G. H. Fecher, J. Kübler, C. Felser, *Materials* 15(17), 5812 (2022)). Then, one can generally indicate as chiral metal an electronic state that has a well-defined handedness, due to the lack of inversion, mirror or other roto-inversion symmetries, while preserving translation and rotation symmetries of the crystal lattice. Starting from this description, in our work we introduce the concept of surface spin-orbital chiral metal to refer to a conducting electronic state of matter that has a well-defined handedness due to an interaction driven magnetochiral order that lacks mirror symmetries and owes the same translation symmetry of the lattice. Indeed, for surface states both inversion and roto-inversion symmetries are naturally broken. In particular, a hallmark of the proposed surface magnetochiral state is that the time-reversal symmetry breaking is due to electronic currents that are chiral due to the fully antisymmetric correlations between the electron spin and orbital moments. We have revised the manuscript in the introduction and worked out the concept of chiral metal and magnetochiral metal along the above discussion in a way that is accessible to a non-expert reader.

With regard to the surface reconstruction, we have considered the electronic structure in the presence of surface reconstruction due to the staggered rotations of the octahedra. While the performed analysis shows that the surface reconstruction does not substantially affect the main conclusions of our study, it allows us to understand in a deeper way the fundamental difference between a chiral

metal and a magnetochiral metal state as well as to address the role of spatially inhomogeneous chiral currents.

Additionally, we have clarified the connection between the measured quantity in the ARPES experiment and the observables evaluated in the model. Below, we report the details of these outcomes in the reply to the point-by-point questions and suggestions of the referee.

Here come some point-by-point questions and suggestions to improve the manuscript in the order as they occur in the text:

1. In the abstract, the authors write about “develop a new theory”. Indeed, there is a microscopic Hamiltonian and the calculation of the angular momentum operator presented in the work. However, the direct link between the electronic structure exhibiting loop currents and the experimental measurements, i.e. the coupling of light to the electronic states and the expected measured quantities could be presented more clearly.

We thank the reviewer for pointing this out. We have revised the manuscript by including more details about the coupling of light to the electronic states as well as the link with the measured quantities. In particular, as expanded below and also reported in the supplementary information, we have reported the basic steps to get from the coupling of the light to the electronic states the expected measured quantities.

2. The authors stress in the abstract that the data is compatible with “quadrupole currents at the surface”. Is there any experimental evidence (from the current study) that the currents only exist on the surface? Or is it merely an indirect conjecture because bulk sensitive measurements have so far not detected any evidence of these currents yet.

We thank the referee for this question. In the present case, the magnetism is known to be surface-derived, thus we just focused on the surface. In general this technique can be truly considered to be surface-sensitive because at the energy used for the study we are probing just the first few layers of the materials. Ideally, to get a true bulk sensitivity, either we need to go down into the deep-UV range (< 10 eV), which would make in the present case impossible to detect the bands because the k -range acceptance becomes too small to see the states, or we need to go to much higher energies (there are currently no facilities to the best of our knowledge to do this). Nevertheless, this study focuses on the surface physics.

Regarding the bulk sensitive measurements, although there are no direct evidence of these currents, we would like to mention that bulk muons under uniaxial strain have found evidence of a magnetic phase that sets in above a critical strain strength. We argue that the application of the strain by reducing the crystalline symmetry from tetragonal to orthorhombic can induce the formation of bulk currents in analogy to what has been observed on the surface. The possibility to measure spin resolved dichroism for the bulk states upon the application of strain is however out of reach for the available experimental probes.

3. page 3, lines 50-70: The authors try to describe the physical consequences of spin and orbital textures in the electronic structure. However, there are a number of concepts introduced and used that would make it difficult for a non-expert reader to understand the essence. The wording could be expanded and made more accessible by illustrating with (known) examples and/or a mathematical description of symmetry properties might be needed.

We are grateful to the referee for these valuable suggestions. We have expanded and revised the

text to make the discussion more accessible to non-expert.

4. page 4, line 85: Similar comment here. What does “profile of LS quadrupole texture” mean here? Indeed, the abbreviation LS is defined, but that quantity should be described better for example by defining it mathematically.

We thank the referee for this suggestion. We have clarified the meaning of quadrupole and quadrupole texture as well as the meaning of the *LS* abbreviation. In general, the product of the spin and orbital angular momentum components can set out a tensor. Then, since the spin and orbital angular momentum are pseudovectors and have a magnetic dipolar character we dub their product as a spin-orbital quadrupole. The referee is right that the meaning of the texting “LS quadrupole texture” was not sufficiently clear and it does not specify the component of the quadrupole tensor that we refer to. Hence, we have clarified the correspondence of the *LS* texture abbreviation with the expectation value of the components of the spin-orbital quadrupole $\hat{L}_i \hat{S}_m$ for a given electronic state and momentum in the Brillouin zone. While the tensor structure of the *LS* product contains several components, the focus is on the $\hat{L}_z \hat{S}_z$ term because it is directly related to the amplitude of the spin-dichroic signal while the antisymmetric components $\varepsilon_{ijk} \hat{L}_j \hat{S}_k$ are relevant to set out the chiral current state.

5. page 4, line 91: The authors use the experimental method of CP-Spin ARPES. The referee believes that some additional discussion is needed to make the reader understand how the spin and orbital moment can be detected with this method.

We thank the referee for bringing out this point. We agree and we have now added in the discussion a new part which is aimed at showing how circular dichroism ARPES is a gold standard technique to detect the orbital angular momentum and how, by adding the spin-species, the spin resolved orbital moment too gets revealed. Moreover, in the supplementary information we have added a brief derivation of the spin-resolved CP-ARPES intensity.

6. Fig. 1: panel (a) Is there a typo for the “Spin” example? Should it read $\langle S_i \rangle \neq 0$ instead of $\langle S_{ij} \rangle$?

We thank the referee for noticing this typo. The referee is correct and there is a typo for the spin example. It should be $\langle S_i \rangle \neq 0$ instead of $\langle S_{ij} \rangle$.

What about the possibility of “spin orbital”, i.e. $\langle S_i \rangle \neq 0$ and $\langle L_i \rangle \neq 0$ (but not the quadrupole moment? Can such a current be created? Does it have distinct properties from just a superposition of spin and orbital currents?

We thank the referee for this observation. The only terms that are not superposition of spin and orbital currents and can lead to $\langle S_i \rangle \neq 0$ and $\langle L_i \rangle \neq 0$ without a quadrupole moment are those related to magnetic interactions in the spin and orbital channels. For instance, the decoupling of Heisenberg interactions (i.e. $\sim J_{l,m} \mathbf{S}(l) \cdot \mathbf{S}(m)$ with nearest neighbor l, m sites) in the spin or orbital channels can lead to terms that are even parity in space (i.e. proportional to a constant value or having a $\cos(k_i)$ factor form) and linear in the spin and orbital angular momentum. This type of interactions and the resulting broken symmetry states, however, are not related to the presence of electronic currents but rather to the occurrence of a spatial ordering of spin and orbital moments.

7. Figure 4 (a): As understood, this is a plot of the spectral function for the surface electronic structure. In detail, this looks quite different from (a) other theoretical models and (b) other

ARPES results on SRO for example the work by Tamai et al. (Ref. 38) Fig. 7 (right). Please explain.

We thank the referee for this observation. We have revised the values of the electronic parameters of the theoretical model to better match the electronic structure with the experimental data. As one can see now from the comparison of Figure 3 and Figure 4 the electronic structure reproduces very well the experimental data and it is compatible with the other ARPES results mentioned by the referee.

8. Figure 4 (b): Can you please explain the significance / meaning of this panel. Is this supposed to be a real space sketch of the currents. If yes, why are the currents running along the $(1,1,0)$ direction and not along $(1,0,0)$ for example as it would be expected for currents carried by the nearest neighbor hopping. How should one understand these currents? Are these supposed to be loop currents that close with a length scale? (What would be this length scale?)

We thank the referee for this observation. We confirm that the sketch in Fig. 4 is a real space cartoon of the current pattern. There are few observations to be made about the origin of these electronic currents and their distribution. First, we would like to point out that, as also observed by the referee, there can be other electronic currents that occur for the Ru-Ru bonds along the $[1,0,0]$ or $[0,1,0]$ directions as well as for the other diagonal $[-1,1,0]$. The contributions along the $[1,0,0]$ or $[0,1,0]$ can manifest as a projection of those along the diagonals or they can be intrinsically driven by the inter-site Coulomb interaction (see below about the origin of the Coulomb induced current phase). However, independently of the current configuration, our conclusions are not qualitatively affected by the presence of other electronic current terms. We have selected the path along the $[1,1,0]$ direction because the momentum scan is along that direction in the Brillouin zone and the ARPES anomalies are observed there. Regarding the fact that the currents along the $[1,0,0]$ or $[0,1,0]$ are smaller in amplitude as compared to those along the $[1,1,0]$ direction, we argue that this circumstance can be due to the fact that the effective Ru-Ru Coulomb interaction along $[1,0,0]$ or $[0,1,0]$ is screened by the presence of the oxygens and by the fact that the kinetic energy is large along those crystal directions. The lack of connecting oxygens along the $[1,1,0]$ direction and the small hybridization among the d -orbitals may result in a larger inter-site Coulomb interaction.

Another reason to have a larger current amplitude for the $[1,1,0]$ Ru-Ru bonds as compared to the $[1,0,0]$ or $[0,1,0]$ directions can be extracted by considering the loop currents resulting from the coupling between ruthenium and oxygen states. Indeed, there can be configurations of intra-octahedra loop currents (as those showed in Fig. 2 c) where the current pattern on the Ru-O_x-Ru bond has opposite orientations on the Ru_A-O_x and O_x-Ru_B bonds. Instead, there are nondestructive current paths, involving the spin and orbital degrees of freedom, that link the ruthenium atoms along the $[1,1,0]$ direction. Such argument indicates that by integrating out the high energy oxygen states, for the considered intra-octahedra loop currents one would have a current amplitude that is about vanishing when considering the $[1,0,0]$ or $[0,1,0]$ directions and sizably different from zero along the $[1,1,0]$ path.

We confirm that our findings indicate the presence of significant anomalies in the spin-dichroic asymmetry that are compatible with the occurrence of sizable chiral currents along the $[110]$ or $[-110]$ directions.

Let us now present how the electronic current phase arises as a broken symmetry state with a nonvanishing expectation value of the current operator on the Ru-Ru bond due to the Coulomb interaction and not to the Ru-Ru hopping connectivity. Indeed, in order to demonstrate this point

one needs to introduce the spin-orbital asymmetric bond operator

$$\phi_{\sigma,\sigma'}^{\alpha\beta}(l, m) = i \left(c_{\alpha,\sigma}^\dagger(l) c_{\beta,\sigma'}(m) - c_{\beta,\sigma'}^\dagger(m) c_{\alpha,\sigma}(l) \right) \quad (1)$$

for the $l - m$ bond between two Ru atoms with position identified by the coordinates R_l and R_m . Here, $c_{\alpha,\sigma}(l)(c_{\alpha,\sigma}^\dagger(l))$ are the annihilation (creation) operators associated with an electronic state with α orbital and spin σ at the atomic site R_l . Then, the spin and orbital dependent terms that build up the density-density inter-site Coulomb interaction U_{lm} for a generic $l - m$ bond can be written in the following form

$$U_{lm} n_{\alpha,\sigma}(l) n_{\beta,\sigma'}(m) = -\frac{1}{2} U_{lm} (\phi_{\sigma,\sigma'}^{\alpha\beta}(l, m))^\dagger \phi_{\sigma,\sigma'}^{\alpha\beta}(l, m) + \frac{1}{2} U_{lm} (n_{\alpha,\sigma}(l) + n_{\beta,\sigma'}(m)) \quad (2)$$

where the orbital and spin resolved density operators, $n_{\alpha,\sigma}(l)$, are defined as $n_{\alpha,\sigma}(l) = c_{\alpha,\sigma}^\dagger(l) c_{\alpha,\sigma}(l)$. Hence, by decoupling the quartic term, one can introduce an order parameter associated with the expectation value of $\phi_{\sigma,\sigma'}^{\alpha\beta}(l, m)$ and express the interaction as

$$U_{lm} n_{\alpha,\sigma}(l) n_{\beta,\sigma'}(m) \sim -\frac{1}{2} \left[\langle \phi_{\sigma,\sigma'}^{\alpha\beta}(l, m) \rangle (\phi_{\sigma,\sigma'}^{\alpha\beta}(l, m))^\dagger + h.c. - |\langle \phi_{\sigma,\sigma'}^{\alpha\beta}(l, m) \rangle|^2 \right] \quad (3)$$

where the average value indicates the summation over all the electronic states weighted by the Fermi distribution function. Taking into account the spin-orbital order parameters on the $l - m$ bond, by suitable superposition of the ϕ operators one can construct a bond current order parameter that is given by the expectation value of the following orbital and spin-orbital quadrupole current operators:

$$\begin{aligned} J_{i,j}^o(l, m) &= i \left(\vec{c}^\dagger(l) \hat{L}_i \hat{L}_j \vec{c}(m) - h.c. \right) \\ J_{i,j}^{so}(l, m) &= i \left(\vec{c}^\dagger(l) \hat{L}_i \hat{S}_j \vec{c}(m) - h.c. \right) \end{aligned} \quad (4)$$

with $\vec{c}^\dagger(l) = (c_{xy,\uparrow}^\dagger(l), c_{yz,\uparrow}^\dagger(l), c_{zx,\uparrow}^\dagger(l), c_{xy,\downarrow}^\dagger(l), c_{yz,\downarrow}^\dagger(l), c_{zx,\downarrow}^\dagger(l))$ and $i, j \in \{x, y, z\}$.

9. page 11, line 180: The authors argue to ignore the surface reconstruction which is supposed to go together with the octahedral rotation. Certainly, the octahedral rotations also break a number of mirror symmetries and introduce handedness i.e. chirality in the (static) electronic structure. Why does this not alter the conclusions /analysis? Could this be the origin of the breaking of the symmetry in the ARPES measurements?

We thank the referee for this valuable remark. This is a very important point and it allows us to clarify relevant conceptual aspects of the examined chiral currents phase as well as the role of mirror and time-reversal symmetries in setting out the anomalous findings of the ARPES experiment. The key point about the structural reconstruction is that, although the rotations of the octahedra break a number of mirror symmetries, they cannot provide an explanation of the asymmetry observed in the ARPES. This is because the rotations do not break the time reversal symmetry and thus the orbital and spin-orbital moment evaluated at k and $-k$ must have amplitudes that are constrained by the time-reversal symmetry. This implies that the observables associated with the dichroic and spin-dichroic signals would have zero asymmetry when comparing the amplitudes at k and $-k$ momenta. Thus, while the rotations of the octahedra can imprint weak mirror symmetry

breaking effects because they primarily affect the oxygens and then get transferred into the Ru d -states, close to the Fermi level, they cannot lead to anomalies of physical observables related to the orbital and spin-orbital momentum due to the time-reversal symmetry constraint.

We have verified this scenario by simulating the electronic structure in the presence of staggered rotations of the octahedra and determined the orbital and spin-projected orbital angular momentum along various directions of the Brillouin zone. The outcome confirms that there is no asymmetry for this electronic configuration.

10. page 11, line 187: The authors state that the mentioned expectation values are “related to the dichroic and spin dichroic amplitudes probed by ARPES”. Can this connection be made a little more precise? What does “related” mean here? Can one see this from the coupling to the vector potential? Additional related question: Are also matrix elements of the dipole operator involved which can have nontrivial effects if mirror symmetry of the orbital states (Wannier states for example) is broken as it happens on the surface?

We thank the Referee for this remark. We have included a section in the supplementary info where we have clarified the connection between the expectation values of the orbital moment and spin-resolved orbital moment with the dichroic and spin-dichroic amplitudes probed by ARPES. We report here the main steps of the derivation and the expressions for the dichroic and spin-dichroic amplitudes. Here, we follow the derivation reported in ref. Phys. Rev. B 85, 195401 (2012) and adapt it to the spin-dichroic case too. The starting point is to consider that the circular dichroism signal probed by ARPES can be expressed through the normalized amplitude as

$$D(k) = \frac{\sum_{\sigma} [I_{\sigma}^R - I_{\sigma}^L]}{I_{\sigma}^R + I_{\sigma}^L}. \quad (5)$$

The sum over the final state spin σ indicates the spin-integrated nature of the detection approach. Thus, one can also introduce the spin-resolved dichroic amplitude as given by

$$D_s(k) = \frac{[I_s^R - I_s^L]}{I_s^R + I_s^L}, \quad (6)$$

here for convenience we assume that the spin configuration is along the z -axis and are labeled by $s = +(-)$ to indicate the \uparrow_z (\downarrow_z) spin states, respectively. Similar expressions can be derived for the all the other spin orientations.

To calculate the amplitude of the dichroic and spin-dichroic signal one has to evaluate the transition probability for an optical excitation between the initial and final states. The interaction with the photon is given by $H_{int} = \mathbf{A} \cdot \mathbf{p}$. The vector potentials are $\mathbf{A} = \frac{\epsilon_1 + i\epsilon_2}{\sqrt{2}}$ for right circularly polarized photons, and $\mathbf{A}^* = \frac{\epsilon_1 - i\epsilon_2}{\sqrt{2}}$ for left circularly polarized photons. Hence, the vector $\mathbf{A} \times \mathbf{A}^* = -i\hat{\mathbf{k}}_{ph}$ sets out the incident photon direction. For convenience and clarity, we report the main steps and expressions of the derivation to show the link of the ARPES intensity with the orbital and spin-projected angular momentum. Let us consider as initial state the Bloch configuration with momentum k that is given by $|\psi_k^I\rangle = \frac{1}{\sqrt{N}} \sum_{i,\alpha,\sigma} \exp[ik \cdot r_i] u_{\alpha,\sigma}(k) |i, \alpha, \sigma\rangle$, where $|u\rangle = \sum_{\alpha,\sigma} u_{\alpha,\sigma}(k) |i, \alpha, \sigma\rangle$ is the Wannier configuration centered at the site r_i associated with the t_{2g} orbitals labeled by $\alpha = xy, xz, yz$ and spin σ . A plane-wave state is assumed for the final state $|F\rangle = \int \exp[ik_F r] |r\rangle$ (A. Damascelli, Z. Hussain, and Z.-X. Shen, Rev. Mod. Phys. 75, 473 (2003) and S. Moser, Journal of Electron Spectroscopy and Related Phenomena 214, 29, (2017)). We recall

that the expectation value of the orbital angular momentum evaluated on the state $|\psi_k^f\rangle$ is given by $\langle \hat{L} \rangle = i \sum_s \mathbf{u}_s \times \mathbf{u}_s^*(k)$, with $\mathbf{u}_s = (u_{yz,s}, u_{zx,s}, u_{xy,s})$ and the spin projected orbital components are expressed as $\langle \hat{L}(1 + s\hat{\sigma}_z) \rangle = i\mathbf{u}_s(k) \times \mathbf{u}_s^*(k)$ with $s = \pm$ for the up and down spin projections along the z -axis. Following the derivation in ref. Phys. Rev. B 85, 195401 (2012), one can show that the spin resolved transition probability can be generally expressed as

$$D_\sigma = \frac{(\mathbf{A} \times \mathbf{A}^*) \cdot \nabla g_\sigma \times \nabla g_\sigma^*}{[(\mathbf{A} \cdot \nabla g_\sigma)(\mathbf{A}^* \cdot \nabla g_\sigma^*) + (\mathbf{A}^* \cdot \nabla g_\sigma)(\mathbf{A} \cdot \nabla g_\sigma^*)]} \quad (7)$$

with

$$g_\sigma(k_F) = \mathbf{u}_\sigma \cdot \nabla_{k_F} f(k_F) \quad (8)$$

where the gradients are respect to the Fermi wave vector k_F , and $f(k)$ is the Fourier transform of the part of the Wannier function that depends only on the radial distance $r - r_i$. The denominator of D is always positive and has a minor role in the dependence of the matrix elements from the orbital angular momentum. The key quantity for our purposes is given by the factor $\Gamma_s = \nabla g_s \times \nabla g_s^*$. One can show ((Phys. Rev. B 85, 195401 (2012))) that the term Γ_s can be expressed as

$$\Gamma_s = \frac{1}{2} \varepsilon^{\alpha\beta\gamma} (\mathbf{u}_s \times \mathbf{u}_s^*)_\alpha \nabla P_\beta f \times \nabla P_\gamma f \quad (9)$$

$$= i \frac{1}{2} \varepsilon^{\alpha\beta\gamma} \langle \hat{L}_\alpha (1 + s\hat{\sigma}_z) \rangle \nabla P_\beta f \times \nabla P_\gamma f \quad (10)$$

with the vector \mathbf{P} having the following components (Q_{yz}, Q_{zx}, Q_{xy}) with $Q_{ij} = \partial_i \partial_j$. We notice that while the structure of the orbital angular momentum is similar to that of p -orbitals because we are using an effective $L = 1$ manifold for the t_{2g} sector, the second order differential operator Q_{ij} takes into account the t_{2g} structure of the orbital configurations. In a similar way, one can show that the spin integrated Γ amplitude is given by

$$\Gamma = i \frac{1}{2} \varepsilon^{\alpha\beta\gamma} \langle \hat{L}_\alpha \rangle \nabla P_\beta f \times \nabla P_\gamma f, \quad (11)$$

with $\varepsilon^{\alpha\beta\gamma}$ the Levi-Civita tensor. Then, one can see that the amplitude Γ and thus the dichroic signal is proportional to the projected orbital angular momentum components $\langle \hat{L}_\alpha \rangle$ with respect to the incident photon direction. Additionally, the spin-dichroic signal is proportional to the spin-projected orbital angular momentum as given $\langle \hat{L}_\alpha (1 + s\hat{\sigma}_z) \rangle$ with $s = \pm$ for projecting spin up and down configurations, respectively. The remaining form factors depend on the Fermi momentum and on the photon energy. Since our study aims to have a qualitative understanding of the anomalies in the transition amplitude probed by Spin CD-ARPES, their contribution does not affect the conclusions of our findings.

In the employed experimental setup the spin orientation is selected in a direction that is perpendicular to the surface (i.e. z) and the incident photon direction is primarily selecting the out-of-plane z -component of the orbital angular momentum. Hence, the dichroic and spin-dichroic signal are proportional to $\langle \hat{L}_z \rangle$ and $\langle \hat{L}_z (1 \pm \hat{\sigma}_z) \rangle$ as considered in the manuscript. We have revised the supplementary information section by inserting the main steps and equations of the above discussion.

11. references: Minor comment. Note that ‘‘Titles of all cited articles are required’’ by stile of the journal. Having the titles already on review stage would make the reading of manuscript and

evaluation of literature easier.

Thanks for the remark, we have accordingly revised the bibliography.

12. Fig. 1 (Supplement): As understood, panel (a) should show the spectral function of the surface states. Apologies for the ignorance here, but it is difficult to see similarities of this spectral function with the theoretically calculated one in Fig. 4 (a). Please comment and point out the important features. Also, the data seems to deviate from the measurements in Ref. 38 and for example Phys. Rev. Lett. 110, 097004 (2013), Chinese Phys. Lett. 29 067401 (2012), Phys. Rev. B 86, 165112 (2012).

We thank the Referee for this remark. We have refined the electronic parameters to better fit experimental outcome. As one can see now from the comparison of Figure 3 and Figure 4 (main text) the electronic structure reproduces quite well the experimental data and it is compatible with other ARPES results on SRO.

Regarding the comparison of our data with those reported in literature, the states observed are not in disagreement, but their intensity is selected differently because the photon energy used and matrix elements were different from our case (see Fig. 7). Indeed, nearly all of the states can be identified, with the only exception of γ' which in this experimental configuration is too weak to be detected.

13. Supplement, page 3, line 56: The authors write “the bulk of Sr_2RuO_4 are unpolarized in spin”. Is this a conclusion from the current experimental data, or a summary from other measurements in the literature. If the latter is true, please indicate that and underline with literature citations.

We apologize, we realized this was not clear. When we normalised the data we used to align the tails which falls on the background, thus, even if the bulk carries finite spin, the background is unpolarized as the spins here are not anymore good quantum numbers and the electrons scatter everywhere randomly. We have now changed this to make it clearer and we stated that the normalization is executed on the background signal. We thank again the referee to made us notice this aspect.

14. Supplement, page 6, eq. (3): Is there any particular significance of writing the components of the normal state Hamiltonian by help of the L_i matrices? This seems just a rewriting of the model in Ref. 1.

We thank the referee for this question. The main reason for writing explicitly the L_i matrices is that it gives a direct access to the symmetry properties of the Hamiltonian especially when the spin-orbital quadrupole currents are included.

15. Supplement, page 7, line 89: A small Rashba spin orbit coupling term is introduced. Indeed, such a term is expected by symmetry and as the authors argue, experimentally this term is bounded to be small. Question: Do the conclusions regarding the calculated expectation values of the spin and orbital moments or other quantities change if this term is set to zero?

We thank the Referee for pointing out the role of the Rashba spin-orbit coupling. In absence of Rashba coupling the electronic states are symmetric under inversion symmetry (I). Then, in the presence of the proposed orbital or spin-orbital quadrupole currents both inversion and time-reversal (T) symmetries are broken but their combination IT is preserved. This implies that the electronic spectrum is doubly degenerate at any k point of the Brillouin zone. For this configuration we have

Figure 7: **Comparison with literature.** Here, we show explicitly the states that we see compared to the literature works mentioned by the referee. Basically, the states observed are not in disagreement, but their intensity is selected differently because the photon energy used and matrix elements were different from our case. Nevertheless, nearly all of the states can be identified, with the only exception of γ' which in this experimental configuration is too weak to be detected.

evaluated the orbital and spin-projected orbital moment for the case of a phase hosting orbital (J^{oo}) or spin-orbital (J^{so}) magnetochiral currents. We find that the orbital moment is identically vanishing while the spin-projected orbital moment is non zero and it can exhibit a symmetric or asymmetric relation when comparing the amplitudes at k and $-k$ depending on the nature of the magnetochiral state. We find that for the case of $J^{so} \neq 0$ the spin-projected orbital moment is symmetric, i.e. $L_z^+(n, k) = -L_z^-(n, -k)$. Instead, for the case of $J^{oo} \neq 0$, the spin-projected orbital moment is asymmetric, i.e. $L_z^+(n, k) \neq -L_z^-(n, -k)$. The presence of surface reconstruction with octahedral rotations does not influence this qualitative behavior. These findings indicate that even in the absence of Rashba coupling the anomaly of the spin-dichroic signal can be accounted by the presence of a chiral orbital quadrupole current state (i.e. $J^{oo} \neq 0$). On the other hand, since inversion is preserved, this electronic configuration cannot yield a nonvanishing orbital moment at any k point and thus the resulting dichroic signal turns out to be identically zero. We thus expect that even for centrosymmetric configurations the anomaly in the spin-CD ARPES can be employed to detect the presence of anomalous magnetochiral phases.

16. General question about coordinate system: Is there some inconsistency between several parts of the manuscript? The experimental data is analyzed along the momentum path $X \rightarrow \Gamma \rightarrow X'$ (see Fig. 3, c). However, the spin and orbital moments are calculated along $M'' \rightarrow \Gamma \rightarrow M$. Please clarify.

We thank the Referee for noticing this mismatch. It is only due to a different choice of the axis label. This mismatch was due to the choice of the Brillouin zone with and without surface reconstruction. In the revised figures we have corrected the notation in a way that all the mpas and orientations are consistent and the theoretical results can be directly related to the experimental data.

Reviewer Reports on the First Revision:

Referees' comments:

Referee #1 (Remarks to the Author):

In my first report I stated strong concerns regarding the validity of the conclusions in the manuscript of Mazzola et al. The authors now present a revised manuscript and a response letter. In the first version of the manuscript the authors claimed that the spin-integrated dichroism is “perfectly” asymmetric when, in fact, the data showed significant deviations from a perfect asymmetry, just like the spin-resolved dichroism. The authors now state that these deviations are close to their “estimated experimental uncertainty” or partly intrinsic. The authors make no efforts to discuss the origin of this substantial “experimental uncertainty” (or to determine it precisely in designated reference measurements), although this would be crucial for judging the reliability of the results and interpretations. The more important questions are, however: in the light of the large uncertainties, what can the authors eventually claim from their data, and does this warrant the strong conclusion of having observed an exotic magnetic state involving spin-orbital loop currents? In my opinion, the answer to the latter question is a clear “no”.

In order to reinforce their arguments, the authors compare the absolute magnitudes of the deviations from a perfect asymmetry for the spin-integrated CD (slightly larger than 10 %) and for the spin-resolved CD (20-30 %). This observation then is claimed to support a weaker version of their previous claim: a relatively larger deviation from a perfect asymmetry in the spin-resolved dichroism, supporting the theory of a spin-orbital moment formation, where the symmetry-breaking of the ordered phase is predominantly apparent only in the combination of L and S.

Even if it were not for the problems discussed further below, this reasoning is just not convincing. Looking at the critical data sets in Fig. 3d, the red curve (spin-up signal) is clearly overestimated in reaching 20 %, a fairer estimate would be 15-16 %. The blue curve reaches its maximum near 30 % only significantly above E_f , i.e. deep in the flank of the spin-resolved intensity peak (Fig. 3ab) which is also affected by the Fermi distribution. It is unclear how this is related to the intensity differences of the peaks shown in Fig. 3a,b and it seems questionable that this feature is related to an intrinsic effect. Thus, even the data sets for one single momentum (why is only one shown, or is Fig. 3d obtained from summing over the three momenta shown in Fig. 3ab?) and one single photon energy barely demonstrate a significantly larger deviation in the spin-resolved signal. Moreover, matrix elements are well known to vary strongly with photon energy, sometimes orders of magnitude within a few eV. It is also well known that the free-electron final state approximation, used in the theory by the authors, does not provide a quantitative description of photoemission intensities, and that the relation between CD and orbital angular momentum is only qualitative (the CD also varies strongly in magnitude with photon energy). In this light, a factor of at best $\sim 2-3$ between the spin-integrated and spin-resolved CD, measured for one single photon energy and one momentum, is not even remotely close to a reliable indicator for the general conclusion that the spin-resolved CD is significantly more pronounced due to a specific property of the initial state.

Note also that that the spin-resolved CD in Fig. 5 of the response letter, which directly corresponds to the data sets in Fig. 3ab only reaches ca. 10 %, which according to the authors is just their experimental uncertainty. This is very concerning since the main arguments of the paper are based on the data in Fig. 3ab.

Geometry and alignment: In my last report I stated that the spin-integrated intensities and CD signals show significant asymmetries with regards to their magnitude and to the peak positions, contradicting the conclusions of the manuscript. The authors attributed the apparent asymmetries in the band peak positions to an artifact of the plotting software and present new data showing symmetric distributions for the intensities. However, also in the revised version the CD momentum distribution curve at E_f in Fig. 2b shows a clear difference in peak positions for positive and negative k . The band dispersion is expected to obey $E(k) = E(-k)$, even in the symmetry-broken state according to the theory in Fig. 4. The asymmetry in the peak positions thus indicates a misalignment and/or errors in the data analysis for the CD measurements. In the supplement the authors estimate a misalignment of 1 deg based on their Fermi surface data. How can they exclude that this misalignment does not lead to appreciable effects? To experimentally exclude misalignment effects it would be necessary to systematically scan the azimuthal orientation across the mirror plane.

The structural surface reconstruction with rotated octahedra breaks mirror symmetry. Since mirror symmetry is already broken explicitly by the atomic structure, deviations from asymmetric CD and spin-resolved CD distributions are expected merely due to the atomic structure and thus cannot be ascribed to an electronically symmetry-broken phase (see also Phys. Rev. Lett. 103, 067005 (2009)).

Additional points:

From a conceptual point of view, it is not clear to me how the present study goes beyond previous works that used spin-resolved photoemission with circularly polarized light to detect magnetically induced mirror- and time-reversal symmetry breaking (see e.g., Schneider C M and Kirschner J 1995 Crit. Rev. Solid State Mater. Sci. 20 179).

Referee #2 (Remarks to the Author):

The authors have revised the manuscript to clarify some parts and also present experimental consolidations of their claims (about sample alignment for example). This is appreciated. As suggested by referees, they have tried to perform temperature dependence of the spin-resolved dichroic signal, but this turns out to be very noisy and it is difficult to use this result to ensure that the effect is linked to the one observed by μ SR.

One of the main problem raised in the referee process (by referee 1 and 2) is whether the detected signal is safely above experimental uncertainty. The authors have honestly compared (now in Fig. 3d) the difference where they expect no (or weaker) signal (grey) and those where they expect signal (red and blue). The signal from spin-resolved dichroism is larger, but by a rather short margin (factor 2-3). What I still find strange is that this signal is shown for k_2 , where the effect is smallest (the k value is not indicated in caption of Fig. 3 and should be added). Why don't they show (at least

in SI) the comparison for the 3 k, especially k1 where the difference is larger in spin-resolved case ?
Could they also show the situation for the MDC of Fig. 2b ?

I am sorry not to be able to give a concluding opinion on this manuscript without this additional check, but I would really like to be sure that the effect is above experimental error beyond reasonable doubt before recommending such a manuscript. It can be argued (as I did) that the experiment is challenging and stimulating, but it is also a tiny effect, difficult to cross check by any other experiments (as it occurs at the surface), with a complicated discussion, so that a very clear and convincing experimental situation is required.

Referee #3 (Remarks to the Author):

The referee has examined the full package of revised manuscript, supplement and detailed answer to the comments from the previous round and acknowledges that the authors have significantly improved the presentation of the findings and the data analysis that leads to the conclusions. As a preamble, let me point out that in the referee process, there were a number of common points raised: These are (1) the question of how novel the method is (2) questions about absence/presence of symmetry in the spin-summed and spin-resolved dichroism and (3) possible other sources of the qualitative behavior of the data that is not related to the spin-orbital current as proposed by the authors.

When coming to these points, it seems that the authors seem not to make much efforts on (1) to reflect earlier literature in the manuscript or supplemental material. Given the constraint on allowed references for Nature, that could point towards a more specialized journal where less constraints are given. Contrary on the question (2) of the bounds of the detected symmetry, the authors now more clearly discuss the data, but at the same time rephrase the conclusions from qualitatively to quantitatively by acknowledging that also in the spin-summed dichroism, a small asymmetry is present that is argued to be only slightly above the experimental resolution while for the spin-resolved one, there is a signal outside the resolution (while at the same time pointing out that the quantity is divided by a small number, thus giving a large statistical variation). The more convincing argument comes from the presentation of data at higher temperatures where the signal becomes not detectable. Still, there is the problem that at elaborated temperatures the resolution decreases and an asymmetry might be overseen in this case. For (3), the authors give detailed bounds on geometrical sources ("alignment") and discuss the additionally present surface reconstruction that indeed could give an asymmetry in the dichroism, but here it is argued that this one is expected to be quantitatively smaller than the measured signal.

In summary, a convincing reply which leaves some space for improvements in reflection of earlier literature and keeps the possibility of misinterpretation of the data in principle while unlikely. The referee believes that the authors have shown the missing pieces of information and believes that this work can be published, whereas few details could be adjusted in a further revision.

1) Introduction of concept of chiral metal; the authors write in line 90, page 4: “while preserving translation and rotation symmetries of the lattice”. Maybe that is just wording, but it seems that the proposed chiral metal state indeed breaks a mirror symmetry while preserving translation, but the rotation symmetry of the lattice seems also to be broken. The referee believes that the preservation of rotation symmetries is not part of the usual definition of a chiral state; also references to some background literature on this topic could be part of the manuscript (and not only in the reply to the referee).

2) Referee 3, point 6, follow-up question: Could the presence of spatial ordering of spin and (independently) orbital moments lead to the similar measurable signal? In other words: Is it possible that a usual spin order in the background of an existing orbital order be another candidate state for Sr₂RuO₄?

3) Referee 3, point 10: Eq. (5) and corresponding equation in SI: It seems that on the r.h.s. there is a missing sum over sigma, see PRB 85, 195401.

4) Referee 3, point 10 continued: After Eq. (8), the authors state that $f(\mathbf{k})$ is the Fourier transform of the radial Wannier function. Does this assume atomic-like Wannier functions which can be written as a product of an angular part and a radial part $f(x,y,z)=Y(\theta,\phi)*f(r)$ where $Y(\theta,\phi)$ are the usual angular harmonics? This would be an approximation that is not fulfilled in the real system since the Wannier functions have tails from hybridization with neighbored atoms and therefore cannot be written as this product. This might be a quantitatively small correction, but could qualitatively change the conclusions about the absence/presence of the symmetry in the dichroism.

Second Report Referee 1

In my first report I stated strong concerns regarding the validity of the conclusions in the manuscript of Mazzola et al. The authors now present a revised manuscript and a response letter. In the first version of the manuscript the authors claimed that the spin-integrated dichroism is “perfectly” asymmetric when, in fact, the data showed significant deviations from a perfect asymmetry, just like the spin-resolved dichroism. The authors now state that these deviations are close to their “estimated experimental uncertainty” or partly intrinsic. The authors make no efforts to discuss the origin of this substantial “experimental uncertainty” (or to determine it precisely in designated reference measurements), although this would be crucial for judging the reliability of the results and interpretations. The more important questions are, however: in the light of the large uncertainties, what can the authors eventually claim from their data, and does this warrant the strong conclusion of having observed an exotic magnetic state involving spin-orbital loop currents? In my opinion, the answer to the latter question is a clear “no”.

We thank the Referee for having analysed our work again. Respectfully, we are in disagreement with the Referee’s conclusions about the quality of our data and our interpretation. As all Referees noticed, this experiment is elegant in uncovering very elusive and subtle effects, such as spin-orbital chiral currents. Having worked hard in the rebuttal, it is surprising and a bit disappointing to read that we made “no efforts to assess the experimental uncertainty”. Instead, contrary to the Referee’s conclusions, we have fairly discussed the experimental uncertainty, exactly in terms of reference measurements, and considered the possibility of having an intrinsic contribution in the dichroic asymmetry. In this version of the manuscript we have made a significant effort to evaluate the uncertainties in a systematic way with the addition of more points. Moreover, we have also considered the occurrence of an intrinsic dichroic asymmetry in the examined spin-orbital chiral phase and other magnetic phases that break mirror and time reversal symmetries. We believe that, as also stated by the other Referees, our conclusions are fair and well corroborated. Below, we recall the main points regarding the experimental uncertainties. Along this line and to further strengthen our conclusions, we point out that we have added the comparison of the dichroic and spin-asymmetry for more k -points.

First of all, as the Referee notices, the source of finite asymmetry of circular-dichroism, might have either intrinsic origin or simply due to the light polarization uncertainty. As we previously discussed, in order to quantify the uncertainty, we performed such measurements on a reference sample. Worth mentioning that these calibration measurements are copious and done on multiple samples (as also are the measurements on Sr_2RuO_4 in the paper, they have been done multiple times on various samples). These measurements indeed, are part of a work itself, now published in Nature Physics - see work by Di Sante et al., Ref. 23 old main text. In this work, it has been used a kagome lattice because at the exact Γ point there is a single well-defined and large gap opened due to the action of solely spin-orbit coupling. In addition, the bands here are also twofold spin-degenerate in this high-symmetry point, being the system non-magnetic. This allowed us to check the asymmetry not only in the circular dichroism signal, but also in the spin-resolved circular dichroism. Therefore, the use of this system was perfect because of these three elements: (1) SOC gap, (2) twofold degeneracy (so one can select single bands with each spin filtering in the VLEED - this would not be possible in single-spin bands, and it would have been more challenging with multi-fold orbital degeneracy), (3) being this at Γ , where all symmetries are conserved. As the Referee notices, in that work (and in ours too), we do not only use the VLEED, but also the ARPES detector in the circular dichroism measurements and the uncertainty (<10%) is the same

in the two cases, corroborating the fact that this is either intrinsic or comes from the light itself (it certainly rules out that something happens to the electrons when scattered against the VLEED targets). It is possible that we did not describe this enough, and we apologize for this. We now added in the supplementary information a more careful description of this procedure.

We still stress that the spin signal is a factor 2 and 3 higher than the uncertainty, which even if small in absolute terms, puts our data confidently out of the error range. Additionally, we emphasize that, as the other Referee (Referee 2) rightfully notices (and we agree with him/her) we should show the spin-asymmetry for various k -points, as well as monitoring that the same behaviour in momentum-distribution curves (MDCs) to give further evidence of the phenomenology. We do this now for all k -points as requested and we do observe always the same value of uncertainty, still, a larger spin-signal in all cases. We agreed in fact with Referee 2 that the effect might be subtle and that all possible checks must be done, and we provided this part too which still agrees with our interpretation.

As the Referee will see, we also collected additional data for MDCs as requested.

In order to reinforce their arguments, the authors compare the absolute magnitudes of the deviations from a perfect asymmetry for the spin-integrated CD (slightly larger than 10 %) and for the spin-resolved CD (20-30 %). This observation then is claimed to support a weaker version of their previous claim: a relatively larger deviation from a perfect asymmetry in the spin-resolved dichroism, supporting the theory of a spin-orbital moment formation, where the symmetry-breaking of the ordered phase is predominantly apparent only in the combination of L and S.

Even if it were not for the problems discussed further below, this reasoning is just not convincing. Looking at the critical data sets in Fig. 3d, the red curve (spin-up signal) is clearly overestimated in reaching 20 %, a fairer estimate would be 15-16 %.

This is not really correct. Aside from the fact that the value is larger than 15-16%, the asymmetry, being the system with broken time-reversal symmetry, can also have one component as large as zero, as long as the other is much larger. We can understand that this was not fully appreciable with only one value of k and now, also thanks to the valuable suggestions of Referee 2, we have shown more k -points. While the uncertainty is still the same for the dichroic amplitude, i.e. 10%, the peaks of the spins are always larger in all cases. This is also reinforced by the fact that the difference stays the same but the single-spin signals change significantly in momentum (see Figures 1 and 2 reported below).

The blue curve reaches its maximum near 30 % only significantly above E_f , i.e. deep in the flank of the spin-resolved intensity peak (Fig. 3 a,b) which is also affected by the Fermi distribution. It is unclear how this is related to the intensity differences of the peaks shown in Fig. 3a,b and it seems questionable that this feature is related to an intrinsic effect. Thus, even the data sets for one single momentum (why is only one shown, or is Fig. 3d obtained from summing over the three momenta shown in Fig. 3a,b?) and one single photon energy barely demonstrate a significantly larger deviation in the spin-resolved signal.

Again, this statement is just wrong, because it simply does not match with the presented results and, thus, we find it very misleading and not fair in respect to our data. As we show below, where we explicitly draw the Fermi level, the maximum of the signal is always either below the Fermi level or, within 12 meV which is the energy resolution, centered on it - certainly not "significantly above" as Referee 1 states, which we found to be a not entirely honest statement. The role of

the Fermi level is important in that it creates a decrease in the electron-density of states dictated by the Fermi-Dirac distribution. However, this is the same for spin-up and spin-down electrons. As the Referee can see in Fig. 1 below, we honestly and it genuinely cannot be affirmed that the maximum signal is above E_F .

Moreover, matrix elements are well known to vary strongly with photon energy, sometimes orders of magnitude within a few eV. It is also well known that the free-electron final state approximation, used in the theory by the authors, does not provide a quantitative description of photoemission intensities, and that the relation between CD and orbital angular momentum is only qualitative (the CD also varies strongly in magnitude with photon energy). In this light, a factor of at best 2-3 between the spin-integrated and spin-resolved CD, measured for one single photon energy and one momentum, is not even remotely close to a reliable indicator for the general conclusion that the spin-resolved CD is significantly more pronounced due to a specific property of the initial state.

For the photon energy, the Referee is totally correct that the photoemission matrix elements change significantly within a few eV. However, as dictated by the Fermi's golden rule, the amplitude $|M_{if}|^2$ entering in the photoemission matrix element (which describes the transition from initial to final states) is independent of the binding energy and is symmetrical in $k_{//}$. This means that, while photon energy is important to understand the absolute value of spin-polarized states, the k-asymmetry in the spin-polarized signal (which is exactly the goal of this work) does not depend on matrix elements. We agree that for quantification purposes, i.e. how strong is the chiral current phase, how strong the surface unconventional magnetism is, one must do this experiment by varying the excitation energy. However, this is beyond our goal, which is fundamental in nature: proving the existence of spin-orbital chiral current (which is independent on the photon energy being quantified by the asymmetry in k - again, as said $|M_{if}|^2$ is $k_{//}$ symmetric).

Note also that that the spin-resolved CD in Fig. 5 of the response letter, which directly corresponds to the data sets in Fig. 3 a,b only reaches ca. 10 %, which according to the authors is just their experimental uncertainty. This is very concerning since the main arguments of the paper are based on the data in Fig. 3ab.

As now already mentioned a few times, we believe that we have clarified this. Also, the fact that the new data shown are all in agreement with our attribution, corroborates (as the Referee 2 suggested) the claim of our work.

Geometry and alignment: In my last report I stated that the spin-integrated intensities and CD signals show significant asymmetries with regards to their magnitude and to the peak positions, contradicting the conclusions of the manuscript. The authors attributed the apparent asymmetries in the band peak positions to an artifact of the plotting software and present new data showing symmetric distributions for the intensities.

This is not fully correct. It is not an artefact of the plotting software: in our previous version, we showed the data in a sheared fashion (to give the sense of 3-dimensionality). Referee 1 was claiming that our standard ARPES data were not symmetric between $+k$ and $-k$, purely by judging them by eye. We believe that this view might have confused the perception of distances (no artefact was introduced), so we decided, also following the suggestions from the other Referees, to show the data in a more traditional way. The latter turned out to be the best choice in fact as the

symmetry is void of doubt.

However, also in the revised version the CD momentum distribution curve at E_f in Fig. 2b shows a clear difference in peak positions for positive and negative k . The band dispersion is expected to obey $E(k) = E(-k)$, even in the symmetry-broken state according to the theory in Fig. 4. The asymmetry in the peak positions thus indicates a misalignment and/or errors in the data analysis for the CD measurements. In the supplement the authors estimate a misalignment of 1 deg based on their Fermi surface data. How can they exclude that this misalignment does not lead to appreciable effects? To experimentally exclude misalignment effects it would be necessary to systematically scan the azimuthal orientation across the mirror plane.

We have to disagree on this point, because it goes all against the experimental evidence and we are sure that this is a factual mistake made by the Referee 1. In our last version (See supplementary information) we explicitly and honestly show the momentum distribution curves at various energies (See also Figure 3 below with MDC obtained by summing the two circular - right and left - signals). The curves do not show any variation at all and the value reported are those from the data. We truly do not understand why the referee is making this statement and we find this a bit unfair given the fact that we reported this not only in a clear way, but also very honestly and explicitly. Maybe this could have been overlooked, but we dedicated a full section in supplementary discussing this. Worth noticing that the values that the referee is talking about when referring to standard ARPES dichroism MDC in Fig. 3 are these (extracted from the software): (1) left peaks; -0.471 and -0.570 $1/\text{\AA}$, and (2) right peaks; 0.475 and 0.573 $1/\text{\AA}$ (and the resolution of the instrument is 0.018 $1/\text{\AA}$). We therefore have to disagree with the Referee 1 and we think that his/her statement is not really fair.

For the azimuth scan, this is tricky, not only because our setup does not have this degree of freedom (currently, to the best of our knowledge, no synchrotron-based infrastructure with a fully commissioned VLEED-twinning detector has this degree of freedom) but also conceptually, as it would be not a very well controlled experiment: as soon as the azimuth deviates quite a bit from the measurement condition, we would not be within the uncertainty in the mirror plane by definition.

The structural surface reconstruction with rotated octahedra breaks mirror symmetry. Since mirror symmetry is already broken explicitly by the atomic structure, deviations from asymmetric CD and spin-resolved CD distributions are expected merely due to the atomic structure and thus cannot be ascribed to an electronically symmetry-broken phase (see also Phys. Rev. Lett. 103, 067005 (2009)).

Again, this aspect has been already touched in our reply and in the revised version of the manuscript (see response to Referee 3). We recall here the main points. This question indeed allows us to clarify relevant conceptual aspects of the examined chiral currents phase as well as the role of mirror and time-reversal symmetries in setting out the anomalous findings of the ARPES experiment. The key point about the structural reconstruction is that, although the rotations of the octahedra break a number of mirror symmetries, they cannot provide an explanation of the asymmetry observed in the ARPES. This is because the rotations do not break the time reversal symmetry and thus the orbital and spin-orbital moment evaluated at k and $-k$ must have amplitudes that are constrained by the time-reversal symmetry. This implies that the observables associated with the dichroic and spin-dichroic signals would have zero asymmetry when comparing

the amplitudes at k and $-k$ momenta. Thus, while the rotations of the octahedra can imprint weak mirror symmetry breaking effects because they primarily affect the oxygens and then get transferred into the Ru d -states, close to the Fermi level, they cannot lead to anomalies of physical observables related to the orbital and spin-orbital momentum due to the time-reversal symmetry constraint.

We have verified this scenario by simulating the electronic structure in the presence of staggered rotations of the octahedra and determined the orbital and spin-projected orbital angular momentum along various directions of the Brillouin zone. The outcome confirms that there is no asymmetry for this electronic configuration (see Fig. 7 and related discussion in the Supplementary Information). There is an important point to be underlined again. We are not discussing whether there is or not a dichroic signal. We are elaborating on the asymmetry of the dichroic and spin-dichroic signal when comparing the corresponding amplitudes at k and $-k$. Our investigations point to a different asymmetry of the dichroic and spin-dichroic signals and the structural reconstruction cannot account for the observed phenomena.

Additional points: From a conceptual point of view, it is not clear to me how the present study goes beyond previous works that used spin-resolved photoemission with circularly polarized light to detect magnetically induced mirror- and time-reversal symmetry breaking (see e.g., Schneider C M and Kirschner J 1995 Crit. Rev. Solid State Mater. Sci. 20 179).

As also elaborated in our previous point, here the spin-resolution is key to achieve the information about spin-orbital chiral phase, therefore the physics discussed in the paper mentioned, despite very interesting, is a different one. However, we also thought that the mentioned work could fit into the general methodology, so we also cite it in the relevant section.

Second Report Referee 2

The authors have revised the manuscript to clarify some parts and also present experimental consolidations of their claims (about sample alignment for example). This is appreciated. As suggested by referees, they have tried to perform temperature dependence of the spin-resolved dichroic signal, but this turns out to be very noisy and it is difficult to use this result to ensure that the effect is linked to the one observed by μ SR.

We thank the Referee for appreciating our efforts. Yes, we indeed tried these measurements (and we even tried them after our second submission) but it turned out that the signal was quite broad.

One of the main problem raised in the referee process (by referee 1 and 2) is whether the detected signal is safely above experimental uncertainty. The authors have honestly compared (now in Fig. 3d) the difference where they expect no (or weaker) signal (grey) and those where they expect signal (red and blue). The signal from spin-resolved dichroism is larger, but by a rather short margin (factor 2-3). What I still find strange is that this signal is shown for k_2 , where the effect is smallest (the k value is not indicated in caption of Fig. 3 and should be added). Why don't they show (at least in SI) the comparison for the $3k$, especially k_1 where the difference is larger in spin-resolved case? Could they also show the situation for the MDC of Fig. 2b?

We are much thankful to the Referee for the fair assessment and for the valuable suggestions. We indeed followed this advice. We apologise we did not show it immediately but we do understand that this does not only give additional confidence to the claim, but it is an indirect way also to monitor the relative level of the spin-integrated dichroism. We have now added this part, which in our opinion, as the referee suggested, corroborates the strength of our results. Additionally, as the Referee suggested we collected also other data which allow us to show the same analysis but for MDC (at two selected energies), and as one can see this is still consistent with our picture and helps us to give further confidence to the results obtained. We did this keeping fixed two binding energies, i.e. at the Fermi level and 150 meV below (which is also where some of the most peaked features appear). We have shown the corresponding outcomes as in the Figures 1,2 below.

I am sorry not to be able to give a concluding opinion on this manuscript without this additional check, but I would really like to be sure that the effect is above experimental error beyond reasonable doubt before recommending such a manuscript. It can be argued (as I did) that the experiment is challenging and stimulating, but it is also a tiny effect, difficult to cross check by any other experiments (as it occurs at the surface), with a complicated discussion, so that a very clear and convincing experimental situation is required.

We thank the referee. We have followed his/her suggestions and accordingly revised the manuscript. We did honestly what the referee was suggesting and we agree that with both EDCs and MDCs analysis the findings are much more convincing. Note that in fact, the spin-independent dichroism value observed in the MDCs, apart for one single k point (our 'unfortunate' initial one) is always well below the error bar and is located around zero. Worth also mentioning that the MDC greatly helped us to draw a confidence region: by integrating the gray line points we found that the spin-integrated circular dichroism is in average 7%, which is also exactly the same value that we estimated in our previous work. Instead the spin-channels, not only are larger, but importantly they vary and show more complexity, for instance they even reverse. This analysis, suggested by the Referee 2 was very useful and gave us further confidence on our claim. We have now added

these figures, with revised caption discussion in the supplementary information.

Second Report Referee 3

The referee has examined the full package of revised manuscript, supplement and detailed answer to the comments from the previous round and acknowledges that the authors have significantly improved the presentation of the findings and the data analysis that leads to the conclusions. As a preamble, let me point out that in the referee process, there were a number of common points raised: These are (1) the question of how novel the method is (2) questions about absence/presence of symmetry in the spin-summed and spin-resolved dichroism and (3) possible other sources of the qualitative behavior of the data that is not related to the spin-orbital current as proposed by the authors.

When coming to these points, it seems that the authors seem not to make much efforts on (1) to reflect earlier literature in the manuscript or supplemental material. Given the constraint on allowed references for Nature, that could point towards a more specialized journal where less constraints are given.

We thank the referee for the remarks. We have revised the reference list to reflect the earlier literature on the methodology. In particular, we have included the following useful references about the methodology: (1) Physical Review Letters 125, 216404, 2020 - (2) Physical Review Letters 110, 216801, 2013 - (3) Physical Review B 85, 195401, 2012 - (4) Scientific Reports 11, 1684 (2021).

Contrary on the question (2) of the bounds of the detected symmetry, the authors now more clearly discuss the data, but at the same time rephrase the conclusions from qualitatively to quantitatively by acknowledging that also in the spin-summed dichroism, a small asymmetry is present that is argued to be only slightly above the experimental resolution while for the spin-resolved one, there is a signal outside the resolution (while at the same time pointing out that the quantity is divided by a small number, thus giving a large statistical variation). The more convincing argument comes from the presentation of data at higher temperatures where the signal becomes not detectable. Still, there is the problem that at elaborated temperatures the resolution decreases and an asymmetry might be overseen in this case. For (3), the authors give detailed bounds on geometrical sources (“alignment”) and discuss the additionally present surface reconstruction that indeed could give an asymmetry in the dichroism, but here it is argued that this one is expected to be quantitatively smaller than the measured signal.

In summary, a convincing reply which leaves some space for improvements in reflection of earlier literature and keeps the possibility of misinterpretation of the data in principle while unlikely. The Referee believes that the authors have shown the missing pieces of information and believes that this work can be published, whereas few details could be adjusted in a further revision.

We thank the Referee for the remarks and for considering the manuscript as worth to be published in Nature. We also appreciate the valuable suggestions to further improve the presentation and content of the manuscript.

1) Introduction of concept of chiral metal; the authors write in line 90, page 4: “while preserving translation and rotation symmetries of the lattice”. Maybe that is just wording, but it seems that the proposed chiral metal state indeed breaks a mirror symmetry while preserving translation, but the rotation symmetry of the lattice seems also to be broken. The Referee believes that the preservation of rotation symmetries is not part of the usual definition of a chiral state; also references to some background literature on this topic could be part of the manuscript (and not only in the reply to the Referee).

We thank the Referee for these valuable observations and remark. A chiral state is usually referred to be a state that lacks symmetries associated with inversion, mirror and roto-inversion transformations (i.e. those transformations whose matrix operators have determinant equal to minus one) (see for instance Moss, G.P. Basic Terminology of Stereochemistry (IUPAC Recommendations 1996) Pure Appl. Chem. 1996, 68, 2193; Flack, H. D. Chiral and achiral crystal structure. Helv. Chim. Acta 86, 905–921 (2003).). Then, the rotation symmetry does not enter directly in setting out the chiral character of the electronic state and it can be either broken or preserved. The Referee is right that the proposed state breaks the C_4 rotation symmetry. The choice of this phase has been dictated by the fact that other experimental observations (i.e. those made by scanning tunneling microscopy, see Ref. 42) have found evidences of rotation symmetry breaking in the spatial dependent patterns of the electronic states at the surface of Sr_2RuO_4 . Hence, for the material upon examination the selected spin-orbital chiral state is more suitable to account for the overall anomalies observed both by muons and by scanning tunneling microscopy. However, one can also construct a spin-orbital chiral state with staggered pattern that preserves C_4 by having spin-orbital quadrupolar current along the other diagonals with rotated $L_i S_j$ quadrupolar components (see for instance Figure 4 below). This state breaks the translational symmetry as it requires a staggering of the fluxes. For such rotational symmetric chiral configuration, the outcome of the dichroic and spin-dichroic amplitude is qualitative the same of that presented in the manuscript. We have revised the text to clarify this aspect of the rotation symmetry breaking. We have also revised the introduction by including few references to background literature for the concept of chiral symmetry.

2) Referee 3, point 6, follow-up question: Could the presence of spatial ordering of spin and (independently) orbital moments lead to the similar measurable signal? In other words: Is it possible that a usual spin order in the background of an existing orbital order be another candidate state for Sr_2RuO_4 ?

We thank the Referee for this observation. We have analyzed several configurations that have both conventional spin and orbital ordering such as to break time reversal and mirror symmetries. For these types of configurations the resulting dichroic and spin-dichroic asymmetries turn out to be always comparable in amplitude. The main point here to support our interpretation is that the spin-orbital chiral current phase is a unique electronic configuration that can exhibit a vanishing dichroic asymmetry while showing a non-zero spin-dichroic asymmetry.

This type of analysis has been indeed performed and reported in the Fig. 9 of the SI. In particular, we have considered the case of a magnetic state with a canted antiferromagnetic configuration in the spin and orbital moments. The broken symmetry state is introduced by an effective magnetic term in the spin channel with uniform/staggered magnetization:

$$H_{\text{mag}} = g\mu_B (\hat{\tau}_0 \mathbf{M}_{\text{uni}} + \hat{\tau}_z \mathbf{M}_{\text{stg}}) \cdot \hat{\sigma}, \quad (1)$$

with \mathbf{M}_{uni} and \mathbf{M}_{stg} being the uniform and staggered magnetization components. Due to the spin-orbit coupling the exchange is also pinning the orbital moment. We have computed, for a representative antiferromagnetic configuration with canted moments, the orbital and spin-resolved orbital moment for all the bands crossing the Fermi level (see Fig. 5 below). As one case see from the inspection of the results in Fig. 5 the orbital and spin-projected orbital moments exhibit a sizable asymmetry when comparing the amplitudes at k and $-k$. This is a general feature of all the magnetic phases with similar symmetry content with respect to mirror and time reversal symmetry

and based on a long-range spatial order of Ru spin-orbital moments.

3) Referee 3, point 10: Eq. (5) and corresponding equation in SI: It seems that on the r.h.s. there is a missing sum over σ , see PRB 85, 195401.

We thank the Referee for the remark. We have corrected the equation in the SI.

4) Referee 3, point 10 continued: After Eq. (8), the authors state that $f(k)$ is the Fourier transform of the radial Wannier function. Does this assume atomic-like Wannier functions which can be written as a product of an angular part and a radial part $f(x,y,z)=Y(\theta,\phi)*f(r)$ where $Y(\theta,\phi)$ are the usual angular harmonics? This would be an approximation that is not fulfilled in the real system since the Wannier functions have tails from hybridization with neighbored atoms and therefore cannot be written as this product. This might be a quantitatively small correction, but could qualitatively change the conclusions about the absence/presence of the symmetry in the dichroism.

We thank the Referee for this valuable remark. We agree with the Referee that the Wannier functions are generally expressed as a superposition of atomic like orbitals. By writing down the complete expression of the Wannier states, one can perform the derivation of the transition amplitude at all orders in the distance from a given atomic center. This correction would lead to contributions containing higher harmonics in the Fermi wave-length λ_F but, in principle, should keep the same functional form. However, independently of the quantitative correction, the impact of this variation can modify the amplitude of the dichroic signal at a given momentum but it cannot introduce an asymmetry between the states at k and $-k$ since this is related to the overall symmetry breaking of the electronic states and it cannot be introduced by the correction of the tail of the Wannier configurations. For this reason, we do not expect that it can change the quality of the results.

Figure 1: **Spin-integrated and spin-resolved dichroism.** Spin integrated circular dichroism collected at **a** $k_y = \pm 0.73 \text{ \AA}^{-1}$ (k_1), **b** $k_y = \pm 0.68 \text{ \AA}^{-1}$ (k_2), and **c** $k_y = \pm 0.62 \text{ \AA}^{-1}$ (k_3), as indicated in the main text Figure 3c. Green curves indicate negative k , orange curve positive k . **d-e-f** Spin-resolved circular dichroism collected at negative k for the three momenta indicated. **g-h-i** Same but collected at positive momenta. **d** The amplitudes of the dichroism (at k -summed up to see the actual residual) are reported and show that while the spin-integrated signal (gray curve in next figure) shows a finite value, as large as 10% (Which is also very similar to the experimental uncertainty as it has shown in Nat. Phys. 19, 1135–1142, 2023 - <https://doi.org/10.1038/s41567-023-02053-z>), the spin-resolved channels show a significantly larger amplitude, of a factor $\times 2$ and $\times 3$ for up and down channels, respectively.

Figure 2: **Amplitude of the dichroism, EDC and MDC.** **a-b-c** The amplitudes of the dichroism (at k -summed up to see the actual residual) are reported for $k_{1,2,3}$. These show that while **d** the spin-integrated signal (gray curve) shows a finite value, as large as 10% (which is also very similar to the experimental uncertainty as shown in Nat. Phys. 19, 1135–1142 2023 - <https://doi.org/10.1038/s41567-023-02053-z> - purple stripe), the spin-resolved channels show a significantly larger amplitude, of a factor larger than $\times 2$ and $\times 3$ for up and down channels, respectively. **e** The amplitude of the dichroism have been also collected by using MDC at two binding energies, i.e., at the Fermi level and at 150 meV below it. As one can see, the gray line, which is the spin-integrated dichroism is nearly flat (in average is 7% - obtained by summing up all the points), while the spin up and spin down channels are varying and well-different. The fact that these are also varying is quite remarkable and indicates that our signal is intrinsic in nature.

List of changes

- Update of references including the link to methods and chiral symmetry.
- Additional analyses about the dichroic and spin-dichroic asymmetry at all k -points as requested by Referee 2.
- Additional measurements to build the MCD as requested by Referee 2.
- Additional discussion about uncertainty and calibration in the Supplementary Information.
- Discussion on the rotational symmetry of the spin-orbital chiral state.
- Correction of typos.

Figure 3: **ARPES identification of the surface states and sample alignment.** **a** Fermi surface collected at 40 eV (sum of the two circularly polarized lights) showing both bulk bands and surface states. The latter are weaker than the bulk in intensity but still visible. To better appreciate the precise sample alignment we fitted the data and extracted the k positions, reported in the image as red markers. The mirror plane deviates from the **(b)** ideal condition by 0.9° . **c**) Energy versus momentum dispersion collected in the same experimental conditions of (Fig.S1a)) showing a very symmetric character. To better appreciate this, we extracted MDCs and plotted them in panel **d** along with their extracted k values.

Figure 4: **Schematic configuration of a C_4 rotational invariant spin-orbital chiral state.** **a** The arrows indicate the current direction on a given bond associated with the corresponding spin-orbital momentum. **b** Schematic representation of a spin-orbital chiral state with C_4 rotation symmetry.

Figure 5: **Orbital and spin-orbital textures in the presence of canted antiferromagnetism and staggered octahedral rotation.** **a** orbital moment $L_z(n, k)$ and **b** spin projected orbital orbital moment $L_z^\pm(n, k)$ for all bands $|\psi_{n,k}\rangle$ evaluated along the $k_y = 0$ direction. The amplitudes of the magnetizations are: $\mathbf{M}_{\text{stg}} = (0, 0, 0.1)$ and $\mathbf{M}_{\text{uni}} = (0.02, 0, 0)$ in units of $\text{eV}(g\mu_B)^{-1}$ (g is the electron g -factor and μ_B is the Bohr magneton).

Reviewer Reports on the Second Revision:

Referees' comments:

Referee #1:

No report

Referee #2 (Remarks to the Author):

In my last report, I asked the authors to demonstrate by additional data that their claim was robust and not biased by the selection of a particularly favorable case. The new data (Fig 5 and 6 in supplementary) convinced me that there is no bias in the choice of the data. (In fact, I got wrongly convinced that the data in Fig. 3d were shown for $|k_2|$, whether it was $|k_1|$. This does not change the conclusion. The k point that is used in Fig. 3d is now indicated on the graph, it could be worth mentioning it again in caption, although it's less critical as everything is shown in supplementary).

This point being clarified, the result presented here remain a subtle effect. I can understand that some people (including referee 1) will still wonder whether it is clear enough to be totally convincing. I still find it interesting to « push » the analysis of CD in ARPES as far as one can to uncover new states of matter. All phenomena related to chiral and orbital order are very elusive, difficult to track experimentally and nonetheless fascinating conceptually. I think this justifies publication of this manuscript in Nature.

Referee #3 (Remarks to the Author):

The authors have again revised the manuscript by adding few explanations and expanded the supplementary information by additional data that underlines the conclusions in the main text. The referee believes that in the rebuttal process, the authors have addressed the core of the critique of raised by the referees. The remaining open question is the debate about the significance of the measured data. As it has been acknowledged by the referees and described by the authors, the experimental setup is challenging to obtain suitable data. Therefore, it turns out that the dichroism when spin-summed reaches the estimated error while the spin-resolved quantity is only a factor 2-3 larger. A fact that the referees suspect to be rather due to systematic errors than an evidence for the chiral currents as the authors advocate. Certainly a difficult question to evaluate and not easy to resolve since systematic errors cannot be captured by data analysis, but only by control experiments which the authors performed in some respect and showed that other quantities as the dichroism MDC's do not exhibit sizable asymmetries as it would be expected from sample misalignment for example. In summary, I believe the data is worth publishing and with the detailed theoretical considerations (now expanded to another possible microscopic picture), sound conclusions are presented.

Still, there is one detail that might need attention to fix:

The authors have added Fig. 5 to the SI (which is also presented in the reply). In the copy-paste process, there might be gotten something mixed up such that the caption of Fig. 5 contains typos of (a) a potentially wrong value of k_y : Should it be 0.72 or 0.62?

(b) mentioning about a “(gray curve)” which is not present in that figure, but probably refers to the “gray curve” in Fig. 6 instead. Please check and correct.

Report Referee 1

The authors provide a revised version of their manuscript in which they added 2 additional data sets of the spin-resolved and spin-integrated dichroism in the supplement. Overall, I still do not think that the present manuscript provides solid (or even any) evidence for the observation of a spin-orbital chiral metal.

The comparison of spin-integrated and spin-resolved asymmetries shows only a factor of 2-3 difference (first the authors claimed a perfectly anti-symmetric spin-integrated dichroism, while in fact the asymmetries are substantial). This difference can readily arise from photoemission effects in a system with symmetry-breaking in the L- and the S-channel (and hence not one with exclusive LS- symmetry breaking, as claimed). The spin-integrated and spin-resolved photoemission dichroic signals also cannot be assumed to quantitatively reflect the intrinsic L and LS character (even if, as the authors seem to argue, a free electron final state approximation suggests so). It is not meaningful to deduce a predominant LS-character, just based on a modest difference in spin-resolved and spin-integrated dichroism.

Respectfully, we are in disagreement with the Referee's conclusions about the quality of our data and our interpretation. As the other Referees noticed, although challenging, the performed experiment is elegant and targeted in uncovering subtle effects of unconventional broken symmetry states, such as spin-orbital chiral currents phases.

We would like to point out that, contrary to the statement of the Referee, the factor 2-3 between the asymmetry of the spin-integrated dichroism and the spin-resolved dichroic amplitude cannot readily arise in a symmetry breaking state for the L- and S-channel. We have indeed considered this aspect during the review process, being also pointed out by the other Referees. To this aim, we recall that in the Supplementary Information we have presented the photoemission effects for a ground-state with antiferromagnetic order whose pattern of spin and orbital moments breaks both time and mirror symmetries in the L- and the S-channel. The resulting orbital and spin-orbital amplitudes, related to the dichroic and spin-dichroic values, respectively, as evaluated at symmetry related momenta are reported in Fig. S9. As one can notice, the asymmetry is sizable in both the dichroic and the spin-dichroic amplitudes and from a general point of view one cannot single out a dominant component with respect to the other when resolving them for band and momentum. For instance, for some of the bands close to the Fermi level and for given windows in momentum space, the spin-dichroic can be even smaller than the dichroic amplitude. Hence, we conclude that this type of broken symmetry states in the L- and the S-channel cannot readily reproduce the trend of the observed asymmetry without a suitable fine tuning in the parameters. Instead, an observation of a small amplitude for the dichroic signal as compared to the spin-dichroic component independently of the momentum and of the selected band can be directly accounted by a ground state hosting spin-orbital chiral currents, as highlighted in Fig. 4 of the main text. This result brought us to conclude that the LS-character of the resulting electronic state is a relevant footprint for understanding the anomalous behavior of the dichroic and spin-dichroic response.

Concerning the correspondence between the dichroic and spin-resolved photoemission signals with the L and LS character of the electronic states, the derivation of the photoemission matrix-element amplitude has been tracked by following both approaches and methodologies which are well established in the scientific community. The section III of the Supplementary Information is fully devoted to this issue.

Finally, for sake of completeness and clarity, we point out that it is not the objective of the

manuscript to achieve an exact quantitative match between experiments and the theory. We also stated this in the main text of the last version, that to achieve something quantitative a fine study similar to ours as function of photon energy and geometries is required. Here, the scope is to show that this methodology can be used to tackle effects compatible with what the theory is predicting.

Regarding the above point I am surprised that the authors state that the matrix element $(M_{if})^2$ is k-symmetric. It is precisely the asymmetry in the (spin-resolved or spin-integrated) matrix elements (or their absolute value squared) that makes up the CD or spin-CD. For example, the CD is $CD = (M_{if})_R^2 - (M_{if})_L^2$, where R and L refer to the helicity of the light.

Frankly speaking we do not understand this criticism. The asymmetry in the polarization and spin resolved matrix elements is the target of the manuscript. The unresolved matrix elements are symmetric in the mirror plane while the asymmetry in the dichroic and spin-dichroic amplitudes have been investigated to get insight into the occurrence of a broken symmetry state with nontrivial coupling of the spin-orbital degrees of freedom.

In addition, I do not have the impression that the authors rigorously rule out instrumental alignment or other inconsistencies, which were already discussed in the last round. In particular, for example, the spin-resolved CD data in Fig. 12 of the supplement only reaches ca. 10 %, which according to the authors is their experimental uncertainty. These are the actual spin-resolved CD curves that correspond to their main data in Fig. 3 of the main manuscript and the difference between these curves is the main finding of the paper. It is unclear to me how the authors reconcile these small spin-resolved CD asymmetries with their estimated uncertainty and their overall claims. Please, also note that in these data sets (Fig. S12), the CD peaks above the Fermi level, which is not expected from the data sets in Fig. 3.

The issue of the instrumental alignment or other setup inconsistencies have been rigorously verified and thoroughly discussed in the section I of the Supplementary Information. When performing this experiment, we reproduced the results on several samples and we did not see any change between these. Within the uncertainty in the azimuth, we could not see any change that affects the results. If this was really significant, we should have at least got significant differences, which we did not.

Furthermore, we believe that when discussing the relation between amplitude and uncertainty of the spin-dichroic the Referee is referring to the Fig. S11 of the supplement and not to the Fig. S12. Apart from that, in Fig. S11 the quantity that is reported is not the spin-resolved CD and thus it cannot be put in correspondence with the data in Fig. 3. The Fig. S11 has been presented to compare the data at low and high temperature. Since the spectra at high-temperature are more fragile than those at low temperature due to the surface degradation, and the signal is reduced quite drastically, within the time allocated we could not get the entire set of measurements as the one at low temperature. We selected quantities, which even if not the same, can be still used to test the presence of an asymmetry in the signal. Then, the comparison of the amplitudes of L and R polarized components with inequivalent spin allowed us to conclude that the strength of the broken symmetry state is suppressed at high-temperature (above the transition temperature that has been indicated by the muons spectroscopy). In the related discussion in the supplement we have clearly addressed the limits of the analysis and the challenges in probing temperature dependent photoemission effects.